# Bad-OOD: Discovering Harmful Synthetic Diffusion Outliers via Confidence Calibration

## Abstract

Utilizing synthetic outlier samples has shown great promise in out-of-distribution (OOD) detection. In particular, impressive results have been achieved by employing diffusion models to generate synthetic outliers in the low-density manifold. However, guiding diffusion models to generate meaningful synthetic outliers remains challenging. The synthesized samples often fall either too close to the in-distribution (ID) data (risking overlap and ambiguity) or too far (leading to visually unrealistic results). Both extremes have been shown to degrade OOD detection performance. In this work, we propose a novel OOD synthesis framework that combines a pre-trained Representation Diffusion Model (RDM) with a simple yet effective classifier calibration strategy. RDM enables global semantic embedding generation without requiring auxiliary labels or text, producing diverse yet ID-relevant outliers, thereby facilitating a more compact ID-OOD decision boundary. To ensure the utility of these samples, we calibrate a binary classifier on both ID data and synthesized OODs to assign confidence-based anomaly scores. We find that mid-confidence outliers, i.e., those balancing realism and deviation, are most informative, and using them significantly boosts detection performance. Extensive experimental results validate the superiority of our calibrated OOD sampler over several strong baselines.

## 1 Introduction

Out-of-distribution (OOD) detection (Liu et al., 2020; Wang et al., 2021; Sun et al., 2022; Huang et al., 2021; Behpour et al., 2023; Sharifi et al., 2024) aims to prevent neural networks from making overconfident predictions when faced with unseen data. This capability is critical for deploying trustworthy machine learning systems in open-world environments, particularly as deep models are increasingly applied to safety-critical domains such as medical diagnostics (Wei & Wang, 2023) and autonomous driving (Liu et al., 2023a).

The core challenge of OOD detection lies in identifying unknown outliers that deviate from the training data distribution. Ideally, an OOD detector functions as a binary classifier that exhibits low uncertainty for in-distribution (ID) data while flagging OOD data as uncertain. A promising recent direction for enhancing OOD detection focuses on synthesizing latent outliers (Lee et al., 2018a; Du et al., 2022; Tao et al., 2023), which can provide explicit supervision and help tighten the decision boundary around the ID manifold.

More recently, Dream-OOD (Du et al., 2023) is proposed to generate visually interpretable OOD images by sampling low-likelihood embeddings in latent space and decoding them through a pixel-space diffusion model. Specifically, it first identifies boundary ID points using KNN distances, then samples outlier embeddings from a Gaussian kernel centered at these points. However, choosing an appropriate sampling radius is non-trivial: samples drawn too far from ID data often result in visually meaningless or unrealistic images, while those too close risk overlapping with ID and confusing the detector. Dream-OOD attempts to balance this by tuning the variance of the Gaussian kernel, but this hyperparameter is difficult to optimize and often dataset- and class-dependent, making the synthesis process fragile. Moreover, the diffusion-based image generation process itself introduces inherent randomness (Ho et al., 2020; Song et al., 2022a; Song et al.), further affecting the reliability of the synthesized outliers. As a result, the above issues can lead to harmful outliers, either resembling covariate-shifted ID or being visually implausible. Despite their importance, these concerns have

been largely underexplored. This raises a key question: *Can we synthesize OOD samples that are anchored to the ID distribution – diverse enough to probe the well-covered boundary, yet reliable in quality – to better separate ID and OOD distributions?*

To achieve the goal, two critical challenges must be overcome: (i) the model should be expressive enough to synthesize diverse yet ID-relevant OOD samples to facilitate a compact decision boundary, and (ii) a robust metric is required to measure the quality of generated outliers, enabling the filtering of harmful samples. To this end, we propose a novel OOD synthesis framework that couples a representation diffusion model (RDM) (Li et al., 2024) with a classifier calibration strategy to jointly address both objectives.

Instead of relying on the conventional diffusion models (Rombach et al., 2022), which sample either in pixel space or instance-level latent space, we adopt the Representation Diffusion Model (RDM) for its ability to generate global semantic representations. This enables the synthesis of diverse outlier embeddings that remain semantically aligned with the ID data. This facilitates the construction of a more compact and robust decision boundary. Moreover, RDM requires no auxiliary class labels or textual descriptions, which are often unavailable, ambiguous, or costly to acquire in real-world settings. In our approach, a self-guided minority metric is applied in conjunction with a pre-trained RDM to generate OOD embeddings, which are subsequently decoded into pixel-space OOD images.

To tackle the second challenge, i.e., evaluating outlier quality, we introduce a simple yet effective classifier calibration strategy. Using both ID data and the full spectrum of synthesized OOD images (including unreliable ones), we train a binary classifier whose prediction probabilities of ID or OOD samples are calibrated to reflect their actual confidence (Guo et al., 2017). Crucially, we observe that the calibrated classifier can already identify harmful OOD samples: low-confidence outliers often resemble covariate-shifted ID samples, while high-confidence ones tend to be visually unrealistic or semantically irrelevant. We show that the performance is improved by retaining the OOD detector using only mid-confidence OOD samples, i.e., those that strike a balance between diversity and relevance. Furthermore, we demonstrate that our calibration strategy is model-agnostic and can be seamlessly integrated into Dream-OOD, leading to substantial gains and highlighting the generality and practicality of our approach.

Our major contributions can be summarized as follows: (i) We extend a self-guided minority metric into latent diffusion space that enables pre-trained RDM to generate diverse ID-relevant synthetic outliers without any auxiliary class labels or text descriptions, which might be difficult to obtain in practice. (ii) A classifier calibration strategy is incorporated into OOD synthesis to measure the anomaly degree of the outliers, thereby preventing harmful OOD data, thus providing more reliable supervision. We demonstrate that the calibrated classifier captures both semantic and covariate shifts by evaluating on Syn-IS (Long et al., 2024), a benchmark designed to disentangle different types of distributional shifts. (iii) Extensive experimental results validate the effectiveness of our proposed OOD calibration method, which outperforms state-of-the-art outlier synthesizers for OOD detection on several benchmarks.

## 2 RELATED WORK

**OOD detection with outlier exposure.** Motivated by the difficulty of acquiring large, high-quality OOD datasets, various techniques have emerged to synthesize auxiliary OOD data that can learn an explicit separation boundary between ID and OOD. Early attempts like Lee et al. (2018a) adopt GAN to generate pixel-level OOD data in a visually interpretable manner. Later, VOS (Du et al., 2022) modeled ID features as a multivariate Gaussian and synthesized outliers by sampling from low-density regions. NPOS (Tao et al., 2023) relaxed the Gaussian assumption by using CLIP (Radford et al., 2021) features to form semantic clusters, and identified sparse feature regions via k-nearest neighbors. Dream-OOD (Du et al., 2023) further bridged feature-space synthesis with visual interpretability by integrating Stable Diffusion (Rombach et al., 2022), enabling image-level realization of outlier features sampled near the ID boundary. Building upon these advancements, we propose a novel *diffusion outlier guidance* strategy that not only retains visual interpretability through diffusion-based generation, but also enables more precise boundary probing without relying on predefined semantic priors. Our method improves the quality and diversity of synthetic outliers, leading to more robust and compact OOD decision boundaries.

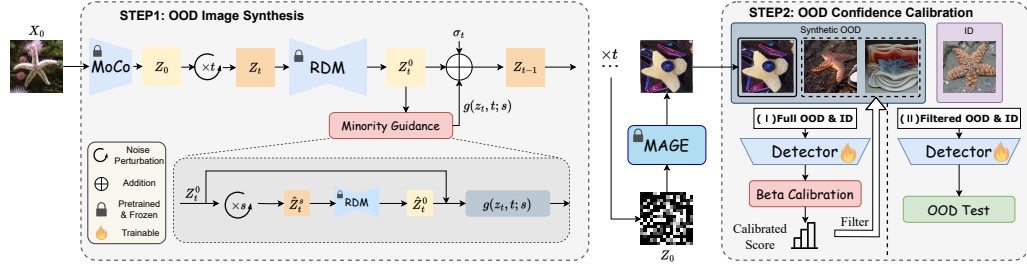

Figure 1: **Overview of our proposed method Bad-OOD.** Bad-OOD first generates auxiliary OOD samples by integrating minority guidance into the denoise process. In the second stage, the detector is initially trained using the entirety of the synthetic OOD samples in conjunction with the ID samples. Subsequent to the beta calibration of the detector, the calibrated scores are employed to filter the previously synthetic OOD samples, thereby excluding those with excessively high or low confidence levels. The refined OOD samples, with the ID samples, are then utilized to retrain the detector from scratch.

**Diffusion models for OOD detection.** Unlike traditional OOD detection methods, recent approaches have leveraged diffusion-based architectures to assess how well a sample aligns with the learned data distribution—typically by reconstructing the input or quantifying deviation during the generative process. Liu et al. (2023b) mask a central region of the image and apply diffusion inpainting, using the restoration quality as an OOD indicator. Gao et al. (2023) combines diffusion inversion with classifier guidance, where high reconstruction error signals a potential OOD sample. A further extension Yang et al. (2024b) adopts multi-layer representations of samples and trains a diffusion model to capture reconstruction error. Additionally, Sun et al. (2022) uses the rate-of-change from a sample toward a standard Gaussian distribution as the OOD score. In contrast to these reconstruction-based approaches, our proposed method is, to our knowledge, the first to directly synthesize feature-level outliers via a diffusion model. This enables flexible, visualizable, and controllable generation of diverse OOD data, which better supports decision boundary learning without requiring reconstruction-based inference.

**OOD detection and distribution shift measurement.** OOD score quantifies how anomalous a sample is concerning the in-distribution (ID) data. MSP (Hendrycks & Gimpel, 2018) utilizes the highest softmax output of a classifier as the OOD score. Despite its simplicity, MSP often suffers from overconfident predictions (Liu et al., 2020), limiting its reliability in estimating OOD severity. ODIN (Liang et al., 2017) improves the separation between ID and OOD samples via temperature scaling and input perturbation. Mahalanobis-based scoring (Lee et al., 2018b) estimates confidence using the distance to the nearest class-conditional Gaussian distribution in feature space. The energy-based method (Liu et al., 2020) mitigates softmax overconfidence by deriving scores directly from the logit outputs. GradNorm (Huang et al., 2021) proposes using the norm of gradients with respect to the input as a signal for OOD detection. KNN-OOD (Sun et al., 2022) utilizes local density differences based on K-nearest neighbor distances between ID and OOD features. Different from prior works, our confidence calibration strategy leverages both reliable and unreliable synthesized outliers to learn an interpretable OOD scoring function. It distinguishes harmful samples from informative ones by aligning prediction confidence with anomaly severity, enabling more effective and targeted supervision for OOD detection.

## 3 OUTLIER SYNTHESIS WITH DIFFUSION MODELS

An overview of the proposed framework is shown in Figure 1, with key components detailed below.

### 3.1 CONDITIONAL ID DATA REPRESENTATION

Our ultimate goal is to generate outliers conditioned on ID data, such that help build a compact boundary around ID data. To achieve it, we can first obtain a representation of the conditional ID data as shown in Figure 1. Specifically, given any ID image $x_0$, we utilize MoCo-v3 (Chen et al., 2021), which is pretrained using self-supervised contrastive learning, as image encoder to map the pixel-space image into the latent feature space. The obtained ID image feature $z_0$ intuitively serves as an anchor point for producing its anomaly variants.

## 3.2 Latent Minority Synthesis using RDM

Given the conditional representation of ID data $z_0$, we employ representation diffusion model (RDM) (Li et al., 2024) to produce outlier embeddings. We choose RDM since it allows us to sample from a representation distribution without providing textual conditions, such as class labels or text descriptions, which might be unavailable or difficult to obtain when synthesizing anomalies. Firstly, we add noise to the conditional representation $z_0$ of each ID image over $t$ time steps to obtain a noisy version $z_t$. Inspired by Um & Ye (2024), we extend the minority metric into latent space based on the denoising timestep $t$ and the posterior mean $\hat{z}_0$, which is obtained via the Tweedie's formula (Efron, 2011) and the pretrained diffusion model. Formally, it is defined as:

$$\mathcal{L}(z_0; t) := \mathbb{E}_{q_{\alpha_t}(z_t|z_0)}[d(z_0, \hat{z}_0(z_t))], \tag{1}$$

where $\hat{z}_0(z_t)$ denotes the model's estimate of the clean latent $z_0$ given the noisy latent $z_t$ at timestep $t$, since the true $z_0$ is only obtained at the end of the reverse process, this intermediate prediction is necessarily used for guiding OOD synthesis. $d(\cdot, \cdot)$ is a distance metric, e.g., L1 or L2, to measure the discrepancy between two features. Intuitively, a higher minority score indicates sample in a low-likelihood region farther from the anchor $z_0$, thus being more abnormal. Therefore, we can derive the sampling towards low-likelihood areas by leveraging the gradient of the minority score. Specifically, for a noisy latent sample $z_t$ at timestep $t$, we first exploit the pretrained RDM $\epsilon_\theta(z_t, t, c)$ to denoise $z_t$ into $z_t^0$, i.e., the predicted clean latent at timestep $t$. We then reintroduce $s$ additional noise steps to obtain $\hat{z}_t^s$, which is denoise again into $\hat{z}_t^0$. According to the minority score mentioned above, our guided gradient $g$ can be obtained by:

$$g(z_t, t; s) := \nabla_{z_t}\mathcal{L}(z_t^0; s) = \nabla_{z_t}\mathbb{E}_{q_{\alpha_s}(\hat{z}_t^s|z_t^0)}[d(z_t^0, \hat{z}_t^0)))]. \tag{2}$$

Lastly, as shown in Figure 1, we follow the standard classifier-guided formulation in ADM (Dhariwal & Nichol, 2021) and use the negative gradient direction indicated by minority guidance to push samples away from dense ID regions, thereby resulting in the outlier representations that naturally occupy the periphery of the ID distribution while retaining coherent structures or semantic cues.

## 3.3 Outlier Image Generator

Simialr to Dream-OOD, we explicitly generate outliers in pixel space, yielding visually interpretable samples, preserving fine-grained details such as texture and color, and aligns more naturally with downstream tasks that operate directly on images. Specifically, MAGE (Li et al., 2023) is leveraged as the image generator to convert the outlier embeddings in latent space into images. During inference, a fully masked image serves as the input of generator, and MAGE reconstructs an output guided by the outlier embeddings.

## 3.4 Energy-based OOD Regularizer

In order to utilize synthetic OOD samples for the separation of ID and OOD distributions, we apply the energy-based uncertainty regularization loss following Du et al. (2022). Specifically, assume the OOD detector's output logit of the n-th ID class is $f_n(\mathbf{x}; \theta)$, then the energy function can be formulated as the LogSumExp operation among all these N logits:

$$E(\mathbf{x}; \theta) := -\log \sum_{n=1}^{N} e^{f_n(\mathbf{x}; \theta)}. \tag{3}$$

Then the uncertainty loss can be defined using a binary sigmoid loss based on the energy function, and the threshold for distinguishing ID and OOD should be set to zero:

$$\mathcal{L}_{\text{uncertainty}} := \mathbb{E}_{\mathbf{v} \sim \mathcal{OOD}}\left[-\log \frac{1}{1 + \exp^{-\phi(E(\mathbf{v};\theta))}}\right] + \mathbb{E}_{\mathbf{x} \sim \mathcal{ID}}\left[-\log \frac{\exp^{-\phi(E(\mathbf{x};\theta))}}{1 + \exp^{-\phi(E(\mathbf{x};\theta))}}\right], \tag{4}$$

where $\phi$ denotes a nonlinear MLP. The overall training objective is to optimize the cross entropy for multi-class classification within ID data coupled with the uncertainty regularizer:

$$\mathcal{L}_{all} = \mathcal{L}_{\text{CE}} + \beta \cdot \mathcal{L}_{\text{uncertainty}}, \tag{5}$$

where $\beta$ is to balance the effect of the regularization term. Once obtained the classifier, we can measure the anomaly degree of OOD samples by the sigmoid of the energy function $E(\mathbf{x}; \theta)$. Formally, we obtain:

$$S(\mathbf{x}; \theta) := \sigma(E(\mathbf{x}; \theta)) = \frac{1}{1 + e^{-E(\mathbf{x};\theta)}}. \tag{6}$$

### 3.5 CONFIDENCE CALIBRATION

Since the existence of bad samples within the synthetic OOD, it is vital to establish an effective mechanism to quantify the degree of anomaly in OOD samples. Therefore, we propose a calibration refined strategy, which leverages a calibrated detector to score samples from an auxiliary dataset and systematically selects high-quality data. Essentially, calibration aims to align the confidence output by the classifier with the actual probability, and the confidence output by a perfectly calibrated detector can reflect the degree of anomaly. Therefore, we introduce beta calibration (Kull et al., 2017), which is not only applicable to sigmoid-based output score but also capable of fitting a wide range of probability distributions. Specifically, given an uncalibrated classifier $\mathcal{C}$ and an auxiliary calibration dataset $\mathcal{D}$, beta calibration learns a map function $\mu_{beta}$ that transforms the raw scores $S$ of samples from $\mathcal{D}$ into new scores $S_c$, ensuring the new scores $S_c$ closely approximate the actual probabilities within $\mathcal{D}$. The formula for the mapping function follows:

$$\mu_{beta}(s; a, b, c) = \frac{1}{1 + 1/\left(e^c \frac{s^a}{(1-s)^b}\right)}, \tag{7}$$

where the parameters $a$, $b$ and $c$ are learnable with $a, b \geq 0$. Practically, we perform calibration using a hold-out validation set consisting of 10% of the ImageNet-100 samples as the ID data and ImageNet-OOD dataset for OOD (see Appendix C.2 for more implementation details). Unlike simpler methods such as temperature (Liang et al., 2017) or Platt scaling (Berta et al., 2024), beta calibration introduces parameters (a, b, c in Eq.(7)) that can better capture skewness and heavy tails, which are typical in the asymmetric, long-tailed score distributions of generated samples. Please refer to Appendix E for more analysis of adopting beta calibration. Once the detector has been calibrated, it can be used to evaluate the anomaly degree of the synthetic OOD samples, which is formulated as:

$$S_c(\mathbf{x}; \theta) = \mu_{beta}(S(\mathbf{x}; \theta); a, b, c). \tag{8}$$

## 4 EXPERIMENTS

In this section, we first compare the OOD detection performance of our approach with several strong competitors. Then, we provide a thorough analysis about the effectiveness and properties of our proposed confidence calibrated OOD detector.

### 4.1 EXPERIMENTAL SETUP

**Dataset.** To evaluate OOD detection performance, we follow the experimental settings of Tao et al. (2023); Du et al. (2023) and use ImageNet-100 as the ID dataset, which is carefully selected from the original ImageNet-1k dataset (Deng et al., 2009). Additionally, we adopt the same OOD test datasets as in Huang & Li (2021); Tao et al. (2023); Du et al. (2023), including subsets of iNaturalist (Horn et al., 2018), SUN (Xiao et al., 2010), Places (Zhou et al., 2018), and Textures (Cimpoi et al., 2013). We also follow the common practice of adopting CIFAR-100 as the ID data for training as well. More details are provided in the Appendix A.1.

**Implementation details.** We implement our framework with PyTorch. MoCo-v3 ViT-L is utilized for extracting representation of ID data, then representation diffusion model (RDM) is employed to generate the synthetic embeddings of outliers, and MAGE-L is adopted for representation-conditioned image generation to decode the outlier embeddings into pixel space. During outlier embedding synthesis, we adopt the DDIM (Song et al., 2022a) sampler to predict the denoised version ($\mathbf{z}_t^0$) of the noisy latent of ID data $\mathbf{z}_t$ (we empirically set $t = 300$). To obtain the gradient of the minority metric, we further perturb $\mathbf{z}_t^0$ with $s = 125$ forward DDPM steps. Then given a specific minority guidance scale, we generate 1,300 OOD samples per class, resulting in a total of 130,000 synthetic outlier images. To train the OOD detector, we employ ResNet-34 as the default backbone and train it for 100 epochs using 100k calibrated mid-confidence OOD samples from scratch. We set the learning rate scheduler with 5 epochs of linear warmup followed by cosine annealing, a momentum of 0.9, a weight decay of 5e-4, an initial learning rate of 0.1, and a batch size of 160. The uncertainty regularization parameter $\beta$ is set to 2.0. More details are provided in the Appendix B and C.

**Competitors.** We compare several strong OOD detection baseline methods, categorized into non-synthesis-based and synthesis-based methods. The majority of existing methods fall into the non-synthesis category. Logit-based methods such as MSP (Hendrycks & Gimpel, 2018), ODIN (Liang et al., 2017), GODIN (Hsu et al., 2020), and Energy (Liu et al., 2020) operate directly on the

Table 1: OOD detection results for IMAGENET-100 and CIFAR-100 as the in-distribution data. The bold and underlined numerals denote the optimal and suboptimal results among the synthesis-based methods, respectively. All the results of Bad-OOD were trained from scratch using the calibrated mid-confidence OOD samples.

| Methods | iNATURALIST | | PLACES | | SUN | | TEXTURES | | Average | |
|---|---|---|---|---|---|---|---|---|---|---|
| | FPR95↓ | AUROC↑ | FPR95↓ | AUROC↑ | FPR95↓ | AUROC↑ | FPR95↓ | AUROC↑ | FPR95↓ | AUROC↑ |
| MSP (Hendrycks & Gimpel, 2018) | 31.80 | 94.98 | 47.10 | 90.84 | 47.60 | 90.86 | 65.80 | 83.34 | 48.08 | 90.01 |
| ODIN (Liang et al., 2017) | 24.40 | 95.92 | 50.30 | 90.20 | 44.90 | 91.55 | 61.00 | 81.37 | 45.15 | 89.76 |
| Mahalanobis (Lee et al., 2018b) | 91.60 | 75.16 | 96.70 | 60.87 | 97.40 | 62.23 | 36.50 | 91.43 | 80.55 | 72.42 |
| Energy (Liu et al., 2020) | 32.50 | 94.82 | 50.80 | 90.76 | 47.60 | 91.71 | 63.80 | 80.54 | 48.68 | 89.46 |
| GODIN (Hsu et al., 2020) | 39.90 | 93.94 | 59.70 | 89.20 | 58.70 | 90.65 | 39.90 | 92.71 | 49.55 | 91.628 |
| KNN (Sun et al., 2022) | 28.67 | 95.57 | 65.83 | 88.72 | 58.08 | 90.17 | 12.92 | 90.37 | 41.38 | 91.20 |
| ViM (Wang et al., 2022) | 75.50 | 87.18 | 88.30 | 81.25 | 88.70 | 81.37 | 15.60 | 96.63 | 67.03 | 86.61 |
| ReAct (Sun et al., 2021) | 22.40 | 96.05 | 45.10 | 92.28 | 37.90 | 93.04 | 59.30 | 85.19 | 41.17 | 91.64 |
| DICE (Sun & Li, 2022) | 37.30 | 92.51 | 53.80 | 87.75 | 45.60 | 89.21 | 50.00 | 83.27 | 46.67 | 88.19 |
| GAN (Lee et al., 2018a) | 83.10 | 71.35 | 83.20 | 69.85 | 84.40 | 67.56 | 91.00 | 59.16 | 85.42 | 66.98 |
| VOS (Du et al., 2022) | 43.00 | 93.77 | 47.60 | 91.77 | 39.40 | 93.17 | 66.10 | 81.42 | 49.02 | 90.03 |
| NPOS (Tao et al., 2023) | 53.84 | 86.52 | 59.66 | 83.50 | 53.54 | 87.99 | 8.98 | 98.13 | 44.00 | 89.04 |
| Dream-OOD (Du et al., 2023) | 26.15 | **95.87** | 39.90 | 93.59 | 38.65 | 93.09 | 57.40 | 84.12 | 40.52 | 91.67 |
| w/ Calib. | **25.62** | 95.48 | 37.29 | 93.09 | 36.67 | 92.95 | 54.06 | 86.37 | 38.41 | 91.97 |
| **Bad-OOD** | 26.25 | 95.53 | **32.29** | **93.76** | **34.17** | **93.48** | 53.33 | 86.01 | **36.51** | **92.20** |
| *ResNetv2-101* | | | | | | | | | | |
| RankFeat (Song et al., 2022b) | 41.31 | 91.91 | 39.34 | 90.93 | 29.27 | 94.07 | 37.29 | 91.70 | 36.80 | 92.15 |
| **Bad-OOD** | 25.10 | 95.51 | 32.50 | 93.75 | 32.40 | 93.72 | 52.71 | 86.14 | **35.68** | **92.28** |
| CIFAR-100 as In-distribution | | | | | | | | | | |
| | SVHN | | PLACES | | LSUN | | TEXTURES | | ISUN | |
| Fake it (Mirzaei et al., 2022) | 85.15 | 77.53 | 77.05 | 76.91 | 56.40 | 76.31 | 70.55 | 80.77 | 67.75 | 84.95 |
| Dream-OOD (Du et al., 2023) | 59.00 | 86.86 | 72.45 | 79.98 | 24.55 | 95.33 | 48.10 | 88.04 | 2.15 | 99.21 |
| w/ Calib. | 55.74 | 88.45 | 70.62 | 80.32 | 24.08 | 95.09 | 41.05 | 89.99 | 1.24 | 99.22 |
| SONA (Yoon et al., 2025) | 3.10 | **99.39** | 44.00 | 88.35 | 18.20 | 96.19 | 58.90 | 85.20 | 63.10 | 86.17 |
| **Bad-OOD** | 54.90 | 89.16 | 68.75 | 86.04 | 24.54 | 94.22 | 40.21 | 91.56 | 1.10 | 99.72 |

classifier's output logits, applying post-processing techniques like temperature scaling or energy scores to differentiate between ID and OOD samples. Distance-based methods, including Mahalanobis (Lee et al., 2018b) and KNN-OOD (Sun et al., 2022), leverage statistical or geometric distances in feature space to estimate how far a sample deviates from the training distribution. Feature-based methods such as ViM (Wang et al., 2022), ReAct (Sun et al., 2021), and DICE (Sun & Li, 2022) further exploit internal network representations—e.g., by measuring residuals (ViM), clipping activations (ReAct), or sparsifying weight contributions (DICE)—to improve OOD separability and robustness. We also compare with RankFeat (Song et al., 2022b), which leverages the difference of singular value distributions between ID and OOD features for OOD detection. For synthesis-based methods, OOD generation has evolved from multivariate Gaussians to GANs (Goodfellow et al., 2014), and more recently to diffusion models (Rombach et al., 2022). Dream-OOD (Du et al., 2023) generates outlier description features and employs a pre-trained diffusion model as the image decoder. SONA (Yoon et al., 2025) introduces mutual interference among semantic regions of different categories during the generation process of the diffusion model, ultimately leading to OOD results. Fake it (Mirzaei et al., 2022) utilizes early-stopped SDEs to synthesize near-distribution OOD supplementary data.

**Evaluation metrics.** Three conventional metrics are employed: (i) **FPR95** - Measures the false positive rate of OOD samples when the true positive rate for ID samples is 95%, making it particularly suitable for scenarios where high recall of OOD samples is required. (ii) **AUROC** - The area under the receiver operating characteristic curve, which reflects the detector's ability to distinguish between ID and OOD samples across varying thresholds. (iii) **AUPR** - Area under the precision-recall curve which places greater emphasis on the detector's capability to correctly identify OOD samples.

## 4.2 OOD DETECTION RESULTS

**Quantitative Results.** As shown in Table 1, our method consistently outperforms baseline models across multiple key OOD detection metrics. In particular, (i) it achieves a clear improvement over Dream-OOD (Du et al., 2023), demonstrating the advantage of our RDM-based synthesis and calibration strategy. (ii) Notably, integrating our confidence calibration into Dream-OOD (w/ Calib. in brown) also yields significant performance gains, highlighting the modularity and general applicability of our approach. (iii) We observe a further performance boost when upgrading the backbone from ResNet-34 to ResNet-101, indicating that stronger classifiers can better leverage our synthesized OOD samples. (iv) Similar trends hold when switching the in-distribution dataset from ImageNet-100 to CIFAR-100, confirming the robustness and scalability of our framework

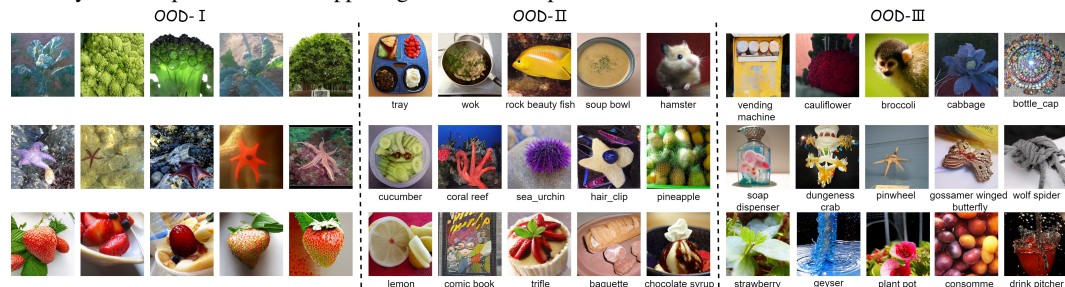

Figure 2: **ID reference samples and synthetic OOD of class broccoli, candle, starfish and strawberry.** The anomaly score is provided in the upper right of each sample.

Figure 3: **Three types of OOD samples of class broccoli, starfish and strawberry.** For OOD-II and OOD-III, the class predicted by CLIP (Radford et al., 2021) is demonstrated under each image.

across varying ID complexities. Due to space limitations, we refer the reader to the Appendix for additional results on synthesis methods and architectures variants (Appendix F), generalization and robustness analysis (Appendix G), similar diffusion-based methods (Appendix H), and fine granularity OpenOOD v1.5 (Zhang et al., 2023) evaluation (Appendix I).

**Qualitative Results.** To further gain some insights about the quality of the generated outlier images, some examples are demonstrated in Figure 2. We can see from the figure that ours can generate high-resolution, semantically meaningful outlier samples similar to Dream-OOD. Uniquely, our method is capable of quantifying the abnormality of each OOD image by estimating its probability of belonging to the ID class, i.e., its classification confidence as ID data. We can figure that a higher score suggests that the OOD sample closely resembles ID data, while a lower score indicates a greater degree of covariance shift, i.e., appearance anomaly. Additional visualizations of synthetic OOD samples alongside their corresponding ID samples are provided in the Appendix J.

## 4.3 MORE ANALYSIS

**Where is bad OOD?** We investigate how our confidence-calibrated binary OOD classifier can locate "toxic" outliers via ablation studies using synthetic OOD samples with varying anomaly levels. Specifically, we verify whether using outliers with either too high or too low OOD scores for training will harm the performance of OOD detector. To reduce randomness, we produce synthetic outliers using multiple minority guidance scales (i.e., 0.025, 0.05, 0.075), yielding 130k synthetic outlier samples (denoted as "Mixed") as OOD candidates. After filtering with our calibrated classifier, we train an OOD detector from scratch with all ID samples (∼ 130k). Results are presented in Table 2. We observe that: (i) Using the full set of 130k synthetic outliers without filtering ("×") yields the worst performance, and randomly

Table 2: Effect with various minority scales.

| Min. Scale | Calib. | FPR95↓ | AUROC↑ | AUPR↑ |
|---|---|---|---|---|
| Mixed | × | 54.06 | 88.75 | 86.30 |
| | Random | 53.52 | 89.43 | 86.86 |
| | High (0.3-1.0) | 51.20 | 89.20 | 86.61 |
| | Mid (0.3-0.5) | 39.69 | 91.71 | 87.13 |
| | Low (0-0.5) | 45.65 | 90.12 | 86.84 |
| 0.025 | Random | 52.01 | 90.00 | 86.87 |
| | ✓ | 41.46 | 91.25 | 86.81 |
| 0.05 | Random | 43.20 | 91.28 | 86.89 |
| | ✓ | 36.51 | 92.20 | 87.32 |
| 0.075 | Random | 44.09 | 91.05 | 86.64 |
| | ✓ | 38.96 | 92.04 | 87.33 |

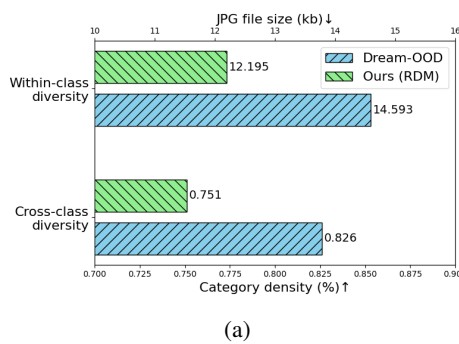 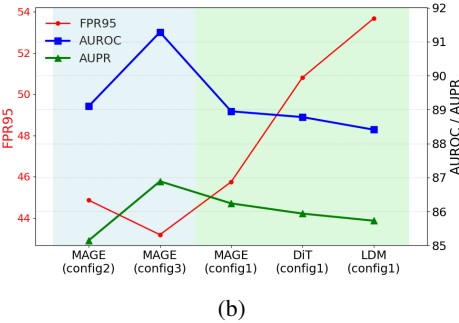

|   |   |
|---|---|
| (a) | (b) |

Figure 4: (a) The diversity of OOD samples generated by our RDM-based method and Dream-OOD. (b) Performance using different image generators.

selecting 100k OOD samples achieves similar results. (ii) The best performance is obtained using mid-level anomaly outliers (OOD scores 0.3–0.5, denoted Mid), likely because these samples are sufficiently distinct yet still near the ID distribution, facilitating a compact decision boundary. (iii) Including near-OOD samples (0.5–1, High) significantly degrades performance (FPR95 increases from 39.69 to 51.20) due to contamination by nearly ID-identical samples; similarly, far-OOD samples provide little benefit as excessively abnormal samples do not aid in learning a compact boundary. (iv) Experiments with specific minority guidance scales further confirm that filtering with the calibrated classifier consistently improves results, demonstrating the robustness of our approach.

**What are the OOD samples?** To examine the content of OOD samples before and after filtering with our confidence-calibrated classifier, we use a pre-trained CLIP image encoder (Radford et al., 2021) to categorize 500 randomly selected synthetic outliers alongside their corresponding ID images. We identify three main types: (i)

Table 3: OOD before and after calibration.

|        | OOD-I       | OOD-II     | OOD-III     |
|--------|-------------|------------|-------------|
| Before | 288 (57.6%) | 45 (9.0%)  | 167 (33.4%) |
| After  | 233 (58.4%) | 38 (9.5%)  | 128 (32.1%) |
| Reduce | 55 (19.1%)  | 7 (15.6%)  | 39 (23.4%)  |

OOD-I: covariance shift outliers with shape/style changes but still recognizable as the original class; (ii) OOD-II: semantic shift outliers, correctly recognized by CLIP but with different class labels; (iii) OOD-III: severe semantic and appearance changes, hardly recognizable even by humans (examples in Figure 3, additional visualizations in Appendix J.2). After filtering with the optimal OOD score threshold (0.3–0.5), most removed samples belong to OOD-I and OOD-III, demonstrating that our method effectively removes near-ID and meaningless far-OOD samples.

**OOD Samples Diversity.** Intuitively, compared to Dream-OOD (Du et al., 2023), our approach replaces class-conditional anchors with ID reference samples for OOD generation, removing the dependency on class labels and enabling sampling from a broader latent space. To quantify the diversity of synthesized OOD samples, we assess both cross-category and within-category diversity. Specifically, we generate OOD samples from 100 randomly selected ID categories, with 100 samples per category. For cross-category diversity, we use a pretrained ResNet-50 classifier to predict labels for all synthesized samples and compute the ratio of unique predicted categories to the total number of ID categories. For within-category diversity, we adopt the metric used in ImageNet (Deng et al., 2009), measuring the average JPEG file size of OOD samples within each category as a proxy for visual variability. As shown in Figure 4a, the RDM produces OOD samples with greater within-category diversity, which in turn encourages a more compact boundary around ID data and improves the generalization ability of the OOD detector.

**Impact of Image Generators.** We further investigate the how different image generators affect the quality of synthesized OOD samples and, consequently, the performance of the OOD detector. As illustrated in Figure 4b, we evaluated three image generator architectures—MAGE (Li et al., 2023), DiT (Peebles & Xie, 2023), and LDM (Rombach et al., 2022)—under the MoCo-B & unconditional RDM configuration (config 1). We can see that MAGE-generated images contribute more effectively to regularizing the OOD detector. We hypothesize that compared to the diffusion-based generator, masked image modeling-based generation demonstrates stronger content understanding, which facilitates the OOD detector's learning process. Additionally, we tested other variants, including MoCo-L & class-conditional RDM (config 2) and MoCo-L & unconditional RDM (config 3), to further assess the role of encoder-decoder combinations.

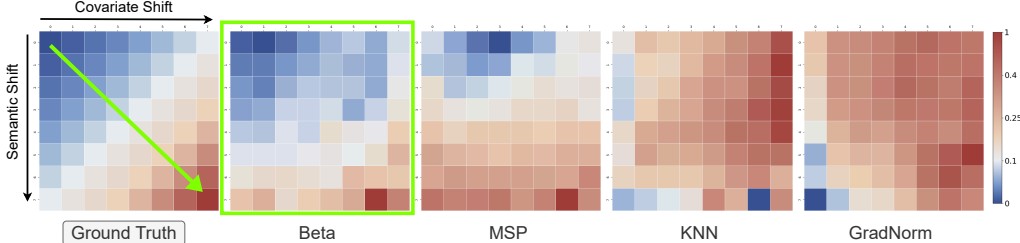

Figure 5: **Comparison of our approach and other competitors.** In each graph, the semantic shift increases from top to bottom, and the covariate shift increases from left to right. All the scores are rescaled to range between 0 and 1, and the color red indicates a higher anomaly degree.

**Computation Cost Analysis.** Since our method samples features at a fixed timestep and filters them with a lightweight confidence model, it is much cheaper than iterative diffusion approaches. The most time-consuming step is image generation via MAGE (single forward pass). On a single Nvidia 3090 GPU, generating 100 OOD ImageNet-100 images takes ∼50s, 2.9s for MoCo feature extraction, 18.2s for RDM perturbation and minority guidance, and 30.3s for latent-to-pixel generation.

### 4.3.1 GENERALIZABILITY ON SEMANTIC AND COVARIATE SHIFT

In this section, we demonstrate that our confidence-calibrated OOD detector effectively captures both semantic and covariate shifts. To this end, we evaluate it on Syn-IS (Long et al., 2024), a benchmark specifically designed to disentangle and quantify different types of distributional shifts, which contains high-quality generated images with more diverse covariate contents.

**Dataset.** SynIS (Long et al., 2024), which systematically quantifies different semantic and covariance shifts relative to the ImageNet-1K dataset, is employed for evaluation. Specifically, images in SynIS are partitioned into several subsets representing eight levels of semantic shift and eight levels of covariance shift, resulting in a total of 64 distinct levels combined semantic and covariance shift for OOD images (More details are provided in the Appendix A.2).

**Competitors.** We compare our approach with several other methods, including: MSP (Hendrycks & Gimpel, 2018), which is a softmax-based score with the higher value indicating the higher probability of being OOD. KNN (Sun et al., 2022) uses the feature distance, with the larger distance indicating the higher probability of being OOD. GradNorm (Huang et al., 2021) adopts gradient information, with the higher back-propagated gradient indicating the higher probability of being OOD.

**Evaluation Metrics.** We calculate the average anomaly scores of the 64 subsets to compare the beta score employed in our method against other classical scoring metrics.

**Results.** As shown in Figure 5, different scoring methods exhibit distinct preferences toward distributional shifts: softmax-based scores such as MSP tend to respond primarily to semantic anomalies while showing low sensitivity to covariate shifts; in contrast, distance-based methods like KNN are more sensitive to covariate shifts but often fail to capture semantic deviations. Gradient-based approaches such as GradNorm struggle to reflect the degree of abnormality in both cases. In comparison, our confidence-calibrated detector more accurately reflects the severity of abnormalities, regardless of whether they arise from semantic or covariate shifts. This improved sensitivity to diverse distributional deviations stems from our minority-guided generation strategy, which promotes a compact and discriminative decision boundary between ID and OOD samples.

## 5 CONCLUSION

In this work, we introduced a novel OOD synthesis approach that integrates a classifier confidence calibration strategy with a pre-trained latent diffusion model. Unlike prior methods, our approach enables controlled sampling in the latent space without relying on auxiliary labels or text descriptions, ensuring the synthesized OOD samples remain both diverse and semantically relevant to the ID distribution. Moreover, we demonstrated that a confidence-calibrated classifier can effectively measure the anomaly degree of synthetic OOD samples, allowing for the removal of harmful outliers that could hinder OOD detection performance. Our findings suggest that utilizing mid-confidence OOD samples leads to a more compact decision boundary and improved detection robustness.

## ETHICS STATEMENT

We are aware of the potential biases in the training data that may lead to the generation of misleading content. Measures have been taken using the method like probability calibration. Overall, our approach is in line with ethical standards, aiming to enhance the reliability and robustness of the model and is reliable for its intended purpose.

## REPRODUCIBILITY STATEMENT

We clarify that the sources of all datasets used in the experiments have been provided in Appendix A. Moreover, the theoretical derivations and properties of the diffusion model and beta calibration have been included in the Appendix B and C, which aids in understanding the rationale for employing these methods. Lastly, the model weights and training code will be made publicly available in due course.

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

# Bad-OOD: Discovering Harmful Diffusion Outliers via Confidence Calibration for OOD Detection (Appendix)

## A    DETAILS OF DATASETS

### A.1    QUANTITATIVE EVALUATION OF OOD DETECTOR PERFORMANCE

We follow the settings of (Tao et al., 2023; Du et al., 2023) to evaluate the performance of OOD detectors, where the classes of OOD test sets do not overlap with ID dataset. For the ImageNet-100 benchmark, we collect subsets from iNaturalist (Horn et al., 2018), SUN (Xiao et al., 2010), Places (Zhou et al., 2018), and Texture (Cimpoi et al., 2013) as large-scale OOD datasets. For the CIFAR-100 (Krizhevsky, 2009) benchmark, SVHN (Netzer et al., 2011), Places (Zhou et al., 2018), LSUN (Yu et al., 2015), iSUN (Xu et al., 2015), and Textures (Cimpoi et al., 2013) are employed as OOD datasets. We provide a brief introduction for each dataset as follows.

**ImageNet-100** is created from ImageNet-1k (Deng et al., 2009) train split by randomly sampling 100 classes, and each class contains approximately 1,300 images with a resolution of 256 × 256.

**CIFAR-100** (Krizhevsky, 2009) comprises 100 fine-grained labels, each containing 600 color images with a resolution of 32 × 32. Among these, 500 images per class are designated for the training set, while 100 images are allocated to the test set. We utilize its training set to assist in training the detector and employ its test set to evaluate the detector.

**iNaturalist** (Horn et al., 2018) contains images of natural world. It has 13 super-categories and 5,089 sub-categories covering plants, insects, birds, mammals, and so on. We use the subset that contains 110 plant classes which do not overlap with ImageNet-1k.

**SUN** (Scene UNderstanding) (Xiao et al., 2010) contains 899 categories that cover more than indoor, urban, and natural places with or without human beings appearing in them. We use the subset which contains 50 natural objects not in ImageNet-1k.

**Places** is a large scene photographs dataset (Zhou et al., 2018), which contains photos that are labeled with scene semantic categories from three macro-classes: Indoor, Nature, and Urban. The subset we use contains 50 categories that are not present in ImageNet-1k.

**Textures** (Describable Textures) (Cimpoi et al., 2013) contains images of textures and abstracted patterns. As no categories overlap with ImageNet-1k, we use the entire dataset as in (Huang & Li, 2021).

**SVHN** (Street View House Numbers) (Netzer et al., 2011) includes 10 classes and each represents a digit from 0 to 9. We use the subset which contains 10,000 samples across all classes.

**LSUN** (Large-scale Scene UNderstanding) (Yu et al., 2015) contains 10 scene categories, such as dining room, bedroom, chicken, outdoor church, and so on. We use the subset which contains 10,000 samples not in CIFAR-100.

**iSUN** (Xu et al., 2015) contains eye tracking ground-truth images derived from the SUN (Xiao et al., 2010) dataset, and we use the entire set of 8926 samples.

### A.2    ADDITIONAL DATASETS USED

In addition to evaluating OOD detection performance, we also introduce the following datasets at the confidence calibration stage and the generalization assessment stage, respectively. Using the ImageNet-OOD (Yang et al., 2024a) dataset, we calibrate our OOD scores with respect to semantic shifts, thereby balancing the model's sensitivity to both semantic and covariate shifts. The detailed calibration procedure is described in Appendix C.2. Additionally, we employ the SynIS (Long et al., 2024) dataset to assess the ability of various OOD scores to capture both types of distributional shifts.

**ImageNet-OOD** (Yang et al., 2024a) is a dataset designed to evaluate the anomaly degree of OOD detection methods purely on semantic shifts, comprising 31,807 images from 637 categories selected directly from ImageNet-21K (Ridnik et al., 2021). It provides a standard test of model performance under semantic shifts.

**SynIS** (Long et al., 2024) consists of 64 subsets, each containing 5,000 images, covering 8 levels of semantic shift and 8 levels of covariate shift, simulating distribution changes by altering semantic content and visual features. This dataset provides diverse distribution shift scenarios for the complement of IS-OOD benchmark, enabling the evaluation of model generalizability under semantic and covariate variations.

# B  PRELIMINARIES

**Diffusion models** (Ho et al., 2020; Song et al., 2022a; Song et al.) are generative frameworks that learn data distributions through a gradual denoising process, which enable controlled generation by incorporating external guidance (e.g., class labels, text prompts) during training and sampling.

Given a data sample $\mathbf{x}_0 \sim q(\mathbf{x}_0)$ and a condition $\mathbf{y}$, the forward process gradually corrupts $\mathbf{x}_0$ by adding Gaussian noise over $T$ timesteps. The noised sample $\mathbf{x}_t$ at timestep $t$ is defined as:

$$q(\mathbf{x}_t|\mathbf{x}_{t-1}) = \mathcal{N}\left(\mathbf{x}_t; \sqrt{1-\beta_t}\mathbf{x}_{t-1}, \beta_t\mathbf{I}\right), \tag{9}$$

where $\beta_t \in (0,1)$ controls the noise schedule. The marginal distribution at timestep $t$ can be expressed in closed form:

$$q(\mathbf{x}_t|\mathbf{x}_0) = \mathcal{N}\left(\mathbf{x}_t; \sqrt{\bar{\alpha}_t}\mathbf{x}_0, (1-\bar{\alpha}_t)\mathbf{I}\right), \tag{10}$$

with $\alpha_t = 1 - \beta_t$ and $\bar{\alpha}_t = \prod_{s=1}^t \alpha_s$.

The reverse process learns to iteratively denoise $\mathbf{x}_t$ while conditioning on $\mathbf{y}$, parameterized by a neural network $\epsilon_\theta$:

$$p_\theta(\mathbf{x}_{t-1}|\mathbf{x}_t, \mathbf{y}) = \mathcal{N}\left(\mathbf{x}_{t-1}; \mu_\theta(\mathbf{x}_t, t, \mathbf{y}), \Sigma_t\mathbf{I}\right), \tag{11}$$

where the mean $\mu_\theta$ is derived from:

$$\mu_\theta(\mathbf{x}_t, t, \mathbf{y}) = \frac{1}{\sqrt{\alpha_t}}\left(\mathbf{x}_t - \frac{\beta_t}{\sqrt{1-\bar{\alpha}_t}}\epsilon_\theta(\mathbf{x}_t, t, \mathbf{y})\right). \tag{12}$$

Lastly the sampling procedure with time dependent constants $\sigma_t$ follows:

$$\mathbf{x}_{t-1} = \mu_\theta(\mathbf{x}_t, t, \mathbf{y}) + \sigma_t\mathbf{z}, \quad \mathbf{z} \sim \mathcal{N}(0, \mathbf{I}). \tag{13}$$

**Logistic calibration.**  A logistic regression classifier is said to be *well-calibrated* if its predicted scores align with the empirical class distribution of the data (Kull et al., 2017). Formally, for a binary classifier predicts scores $s = f(\mathbf{x}) \in [0,1]$, it's required that the proportion of positive instances among those assigned score $s$ equals $s$. Mathematically, this is expressed as:

$$s = \mathbb{E}[Y \mid f(X) = s], \tag{14}$$

where $X$ and $Y$ denote the input sample and binary label of a randomly drawn instance from the dataset. The expectation in equation 14 can be computed empirically as:

$$\mathbb{E}[Y \mid f(X) = s] = \frac{\sum_{j=1}^n y_j \cdot \mathbb{I}[f(\mathbf{x}_j) = s]}{\sum_{j=1}^n \mathbb{I}[f(\mathbf{x}_j) = s]}, \tag{15}$$

where $\mathbb{I}[\cdot]$ is the indicator function, and $n$ is the total number of instances.

For any fixed classifier $f$, there exists a unique calibration map $\mu(s) = \mathbb{E}[Y \mid f(X) = s]$ that achieves perfect calibration on the training data. However, such a map risks severe overfitting, particularly when $f$ assigns unique scores to individual instances. In this case, the calibration map collapses to $\mu(s_i) = y_i$, resulting in overconfident 0/1 predictions that generalize poorly to unseen data. Thus, the goal of calibration methods is to learn a *generalizable* map $\mu(s)$ that preserves ordinality while aligning predictions with empirical frequencies on held-out data.

## C  BETA CALIBRATION

### C.1  DERIVATION OF BETA CALIBRATION.

We provide a core derivation of the beta calibration method (Kull et al., 2017). The key assumption is that the classifier's output scores for each class follow the beta distributions. Therefore, this distribution function can be used to derive the confidence calibration procedure for sigmoid-based binary classification.

**Class-conditional score distributions.**  Assume the scores $s$ for positive ($Y = 1$) and negative ($Y = 0$) instances follow beta distributions with parameters ($\alpha_1, \beta_1$) and ($\alpha_0, \beta_0$), respectively:

$$p(s \mid Y = 1) = \frac{s^{\alpha_1 - 1}(1 - s)^{\beta_1 - 1}}{B(\alpha_1, \beta_1)}, \tag{16}$$

$$p(s \mid Y = 0) = \frac{s^{\alpha_0 - 1}(1 - s)^{\beta_0 - 1}}{B(\alpha_0, \beta_0)}, \tag{17}$$

where $B(\cdot, \cdot)$ is the beta function.

**Likelihood ratio derivation.**  The likelihood ratio (LR) between the two classes is:

$$LR(s) = \frac{p(s \mid Y = 1)}{p(s \mid Y = 0)} = \frac{s^{\alpha_1 - \alpha_0}(1 - s)^{\beta_0 - \beta_1}}{B(\alpha_1, \beta_1)/B(\alpha_0, \beta_0)}. \tag{18}$$

Let $a = \alpha_1 - \alpha_0$, $b = \beta_0 - \beta_1$, and $K = B(\alpha_1, \beta_1)/B(\alpha_0, \beta_0)$. Reparameterizing $K = e^{-c}$, the LR becomes:

$$LR(s; a, b, c) = \frac{s^a}{(1 - s)^b} \cdot e^c. \tag{19}$$

**Calibrated probability mapping.**  The calibrated probability $\mu_{beta}(s)$ is derived from the posterior odds under a uniform class prior:

$$\mu_{\text{beta}}(s; a, b, c) = \frac{1}{1 + LR(s; a, b, c)^{-1}} = \frac{1}{1 + \frac{(1-s)^b}{s^a e^c}}. \tag{20}$$

Simplifying yields the beta calibration family:

$$\mu_{\text{beta}}(s; a, b, c) = \frac{1}{1 + \frac{1}{e^c \cdot \frac{s^a}{(1-s)^b}}}. \tag{21}$$

**Monotonicity constraints.**  To ensure the calibration map is non-decreasing, parameters $a$ and $b$ must satisfy $a, b \geq 0$. This preserves the ordinal relationship between classifier scores and calibrated probabilities.

### C.2  DETAILED IMPLEMENTATION OF BETA CALIBRATION.

We provide a detailed description of the beta calibration method for OOD detectors in Algorithm 1. Specifically, the algorithm takes as input an uncalibrated classifier and calibration datasets, where the ID validation set is drawn from 10% of the held-out data in ImageNet-100, and the OOD validation set is constructed from the ImageNet-OOD (Yang et al., 2024a) dataset. The calibration parameters "abc" are initialized at the beginning of the algorithm. By iterating over both the ID and OOD validation sets, the algorithm computes the pre-calibration probabilities for each sample and fits the calibration parameters according to the sample labels. The final output is a classifier calibrated with beta parameters.

---

**Algorithm 1** Beta Calibration of OOD Detector

---

**Input:** Uncalibrated classifier $\mathcal{F}_\theta$, validation ID dataset $\mathcal{D}_{in}$, validation OOD dataset $\mathcal{D}_{out}$
**Output:** Calibrated classifier $\mathcal{F}'_\theta$
**Initialize** Beta calibration parameters $a_0, b_0, c_0$, empty probability list $P = [\,]$ and label list $Y = [\,]$
**foreach** *mini-batch* $(x_{id}, y_{id})$, $(x_{ood}, y_{ood})$ *in* $\mathcal{D}_{in}$, $\mathcal{D}_{out}$ **do**
  │ **Compute** $p = \mathcal{F}_\theta(\text{concat}(x_{id}, x_{ood}))$
  │ $P \leftarrow p$, $\quad Y \leftarrow \text{concat}(y_{id}, y_{ood})$
**end**
**Fit** beta calibration: $a, b, c \leftarrow \text{fit}(P, Y)$
**Predict** beta calibration: $\hat{P} \leftarrow \text{predict}(P)$
**return** calibrated model $\mathcal{F}'_\theta$ with beta calibration applied

---

# D  PARAMETER SENSITIVITY ANALYSIS.

## D.1  PERTURBATION STEPS $t$.

The parameter analysis on perturbation steps $t$ during OOD synthesis reveals critical insights into the trade-offs between detection performance and model robustness. Notably, $t = 300$ achieves the optimal balance, while further increasing $t$ to 400 introduces a marginal degradation in FPR95 and ID ACC, despite a slight AUPR improvement.

Table 4: Analysis on the perturbation steps during OOD synthesis.

| $t$ | FPR95↓ | AUROC↑ | AUPR↑ | ID ACC↑ |
|---|---|---|---|---|
| 200 | 45.78 | 90.89 | 86.87 | 87.74 |
| 300 | **43.20** | **91.28** | 86.87 | **88.22** |
| 400 | 43.52 | 90.84 | **87.06** | 87.44 |

## D.2  REGULARIZATION WEIGHT $\beta$.

The parameter analysis on regularization weight $\beta$ during OOD detector training shows critical trade-offs between OOD detection robustness and ID classification accuracy. We ablate different choices, as shown in the table below, a moderate value $\beta = 0.2$ achieves the optimal balance.

Table 5: Analysis on the regularization weight during OOD detector training.

| $\beta$ | FPR95↓ | AUROC↑ | AUPR↑ | ID ACC↑ |
|---|---|---|---|---|
| 0.1 | 47.42 | 89.59 | 97.50 | 78.76 |
| 0.2 | **36.51** | **92.20** | **98.08** | **87.44** |
| 0.5 | 50.21 | 89.47 | 97.58 | 79.68 |
| 1.0 | 69.24 | 81.12 | 95.43 | 67.62 |

# E  COMPARISON WITH OTHER CALIBRATION STRATEGIES

We conducted a comparison with temperature scaling (Liang et al., 2017) and isotonic regression (Berta et al., 2024), which are two widely-used calibration techniques. As shown in Table 6, beta calibration consistently yields better AUROC and FPR@95% results across multiple benchmarks, demonstrating its superiority in aligning anomaly scores with the underlying uncertainty of generated OOD data.

Table 6: Comparisons of different calibration strategies with ImageNet-100 as in-distribution data.

| Methods | iNaturalist | | Places | | Sun | | Textures | | Average | |
|---|---|---|---|---|---|---|---|---|---|---|
| | FPR95 | AUROC | FPR95 | AUROC | FPR95 | AUROC | FPR95 | AUROC | FPR95 | AUROC |
| Beta calibration | **26.25** | **95.53** | **32.29** | **93.76** | **34.17** | **93.48** | **53.33** | 86.01 | **36.51** | **92.20** |
| Temperature scaling | 38.85 | 92.50 | 51.04 | 88.34 | 55.73 | 85.56 | 53.65 | 89.10 | 49.82 | 90.38 |
| Isotonic Regression | 41.77 | 90.26 | 53.44 | 86.86 | 55.94 | 83.37 | 54.38 | **89.31** | 51.38 | 88.95 |

# F    COMPARISON OF OUTLIER SYNTHESIS METHODS AND MODEL ARCHITECTURES

In this section, we separately evaluate the impact of outlier embedding generation methods and model architectures. It is important to note that all experiments in this part are conducted under calibrated settings. We first introduce Gaussian noise at each denoising step of RDM (Li et al., 2024) for outlier synthesis and observe that its performance is significantly inferior to that of the minority guidance approach, indicating that the low-likelihood target provided by minority guidance is more explicit. Furthermore, we assess our method on both ResNet-101 (He et al., 2016) and T2T-ViT (Yuan et al., 2021) architectures. Compared to RankFeat (Song et al., 2022b), our results show that the samples selected after calibration contain more effective boundary information than those based on singular value.

Table 7: Comparison of outlier synthesis methods and model architectures.

| Outlier Synthesis | Architecture | FPR95↓ | AUROC↑ | AUPR↑ |
|---|---|---|---|---|
| Add Gaussion Noise | ResNet-34 | 57.14 | 87.24 | 83.89 |
| Minority Guidance | ResNet-34 | 36.51 | 92.20 | 87.32 |
| RankFeat (Song et al., 2022b) | ResNet-101 | 36.80 | 92.15 | - |
| Minority Guidance | ResNet-101 | 35.68 | 92.28 | 87.37 |
| RankFeat (Song et al., 2022b) | T2T-ViT | 51.58 | 85.60 | - |
| Minority Guidance | T2T-ViT | 50.21 | 89.47 | 84.58 |

# G    GENERALIZATION AND ROBUSTNESS ANALYSIS

## G.1    GENERALIZABILITY ON SEMANTIC AND COVARIATE SHIFT.

In this section, we provide more evaluation results of the generalizability of different anomaly measurement approaches and the impact of using different OOD synthesizers. First, we comprehensively present the score performance of more softmax-based methods, distance-based methods, and gradient-based methods on the SynIS (Long et al., 2024) dataset (see Figure 6). In addition, we also assess the previously proposed NPOS (Tao et al., 2023) and Dream-OOD (Du et al., 2023) methods (see Figure 7 & 8).

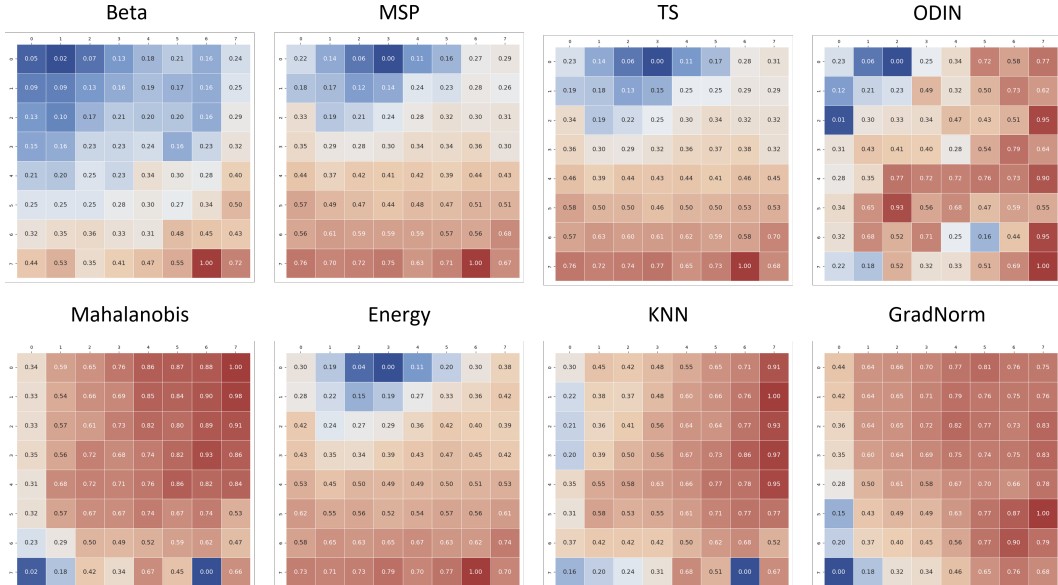

Figure 6: **Full results of beta score and other metrics.** In each graph, the semantic shift increases from top to bottom, and the covariate shift increases from left to right. All the scores are rescaled to range between 0 and 1, and the color red indicates a higher anomaly degree.

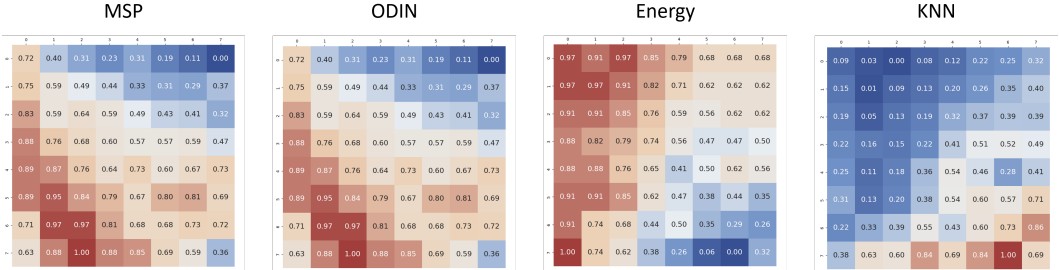

Figure 7: **Full results of SynIS using NPOS (Tao et al., 2023).** In each graph, the semantic shift increases from top to bottom, and the covariate shift increases from left to right. All the scores are rescaled to range between 0 and 1, and the color red indicates a higher anomaly degree.

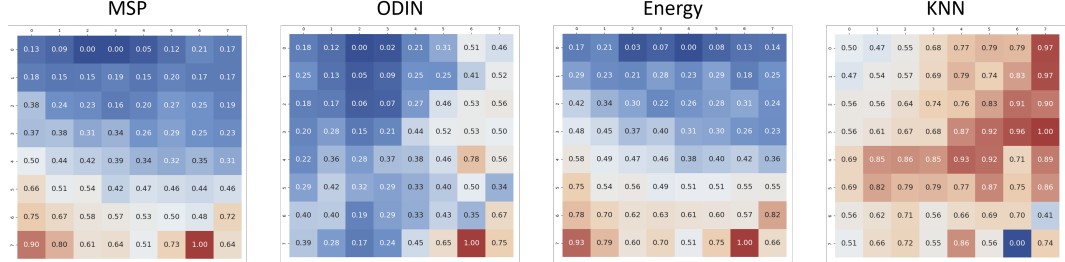

Figure 8: **Full results of SynIS using Dream-OOD (Du et al., 2023).** In each graph, the semantic shift increases from top to bottom, and the covariate shift increases from left to right. All the scores are rescaled to range between 0 and 1, and the color red indicates a higher anomaly degree.

Specifically, given an input image, the classifier generates a prediction, after which different methods compute their respective scores. We provide a detailed introduction to each method as follows.

- Maximum Softmax Probability (Hendrycks & Gimpel, 2018) method uses the largest softmax output of the classifier as the score.

- Temperature Scaling (Liang et al., 2017) method outputs the maximum softmax probability after temperature calibration.

- ODIN introduces (Liang et al., 2017) input preprocessing to enhance the separation between ID and OOD samples, and combines this with temperature scaling to produce the highest softmax score.

- Energy-based (Liu et al., 2020) method replaces the softmax probability with an energy score, addressing the issue of overconfident softmax predictions.

- Mahalanobis Distance (Lee et al., 2018b) method first learns a class-conditional Gaussian distribution on the ID data, and then uses the Mahalanobis distance between the OOD input and the nearest class Gaussian as the anomaly score.

- KNN-OOD (Sun et al., 2022) method computes the K-nearest neighbor distance between the OOD input and ID samples as the score.

- GradNorm (Huang et al., 2021) optimizes the KL divergence between the softmax output and a uniform distribution, and uses the norm of the backpropagated gradient vector as the anomaly score.

## G.2 ROBUSTNESS UNDER INTRA-CLASS AND INTER-CLASS VARIABILITY.

While ImageNet-100 and CIFAR-100 already exhibit a reasonable degree of intra-class variability and inter-class semantic proximity (e.g., fine-grained animal species or visually similar objects), we believe that evaluating under more challenging settings would further strengthen the validation of our framework. To this end, we have conducted additional experiments using in-distribution datasets with increased semantic ambiguity—specifically, 10 manually selected ImageNet-100 classes with high inter-class similarity, and 10 randomly selected ImageNet-R classes with higher intra-class variability. As shown in Table 8, our proposed model consistently outperforms baseline methods, indicating its robustness in these ambiguous scenarios. We believe this robustness stems from RDM's operation in the feature space of ID data, where the diversity-aware generation strategy naturally encourages the synthesis of boundary-sensitive samples, regardless of the inherent class separability.

Table 8: Generalization capacity of our RDM for outlier synthesis beyond standard benchmarks.

| Settings (In-distribution Data) | Ours | | Dream-OOD | |
| --- | --- | --- | --- | --- |
| | FPR95 (Avg.) | AUROC (Avg.) | FPR95 (Avg.) | AUROC (Avg.) |
| 10 Random classes from ImageNet-100 (stingray, jellyfish, Chihuahua, tiger, grasshopper, zebra, accordion, basketball, castle, lipstick) | 66.85 | 77.41 | 68.59 | 76.93 |
| 10 Semantic overlap classes from ImageNet-100 (coyote, tabby, leopard, lion, tiger, zebra, hog, ox, impala, mink) | 68.62 | 79.55 | 78.91 | 69.40 |
| 10 Intra-class variability classes from ImageNet-R (stingray, jellyfish, Chihuahua, tiger, grasshopper, zebra, accordion, basketball, castle, lipstick) | 69.64 | 75.19 | 79.66 | 70.34 |

## G.3 ROBUSTNESS UNDER CLASS-IMBALANCED SETTING.

While our main experiments were conducted on balanced datasets, we further examined the effectiveness of the proposed strategy under class-imbalanced setting by randomly removing a varying number of samples (ranging from 0 to 1200) from each class in ImageNet-100, resulting in a new imbalanced version of the dataset. As shown in Table 9, our minority guidance strategy still yields consistent and noticeable improvements over Dream-OOD under these imbalanced conditions. This empirically suggests that our approach is inherently robust to class imbalance.

Table 9: OOD detection results using the imbalanced version of ImageNet-100 as in-distribution.

| Methods | iNaturalist | | Places | | Sun | | Textures | | Average | |
| --- | --- | --- | --- | --- | --- | --- | --- | --- | --- | --- |
| | FPR95 | AUROC | FPR95 | AUROC | FPR95 | AUROC | FPR95 | AUROC | FPR95 | AUROC |
| Ours | **34.27** | **87.10** | **42.15** | **84.31** | **48.65** | **89.03** | 49.85 | **85.60** | **43.88** | **85.03** |
| Dream-OOD | 50.62 | 82.24 | 50.31 | 80.41 | 59.48 | 76.37 | **46.77** | 78.04 | 54.30 | 79.26 |

## H  COMPARISON WITH SIMILAR DIFFUSION-BASED METHODS

We conducted additional experiments comparing our method against SONA (Yoon et al., 2025) and Fake it till you make it (Mirzaei et al., 2022). The results are summarized in Table 10. We can see that our method achieves better performance in terms of both average FPR95 (37.10) and AUROC (92.97), compared to SONA (FPR95: 37.46 and AUROC: 91.06) and Fake it till you make it (FPR95 at 71.38 and AUROC at 79.29).

Notably, although SONA also utilizes semantic/nuisance disentanglement, our approach differs in two key aspects: (i) Instead of relying on architectural constraints for disentanglement, we employ a classifier-guided calibration that allows flexible and fine-grained control over semantic shift and harmful perturbations; (ii) Our approach is also computationally more efficient. We conducted a dedicated comparison of generation and inference times, showing that our method is 3x faster compared to SONA. This improvement is largely attributed to our use of the lightweight RDM model, which does not require additional conditioning inputs, in contrast to SONA's reliance on SD2 (Rombach et al., 2022), whose joint image-text reasoning introduces higher computational overhead.

Table 10: OOD detection results of diffusion-based methods using CIFAR-100 as in-distribution data.

| Methods | SVHN | | Places | | LSUN | | Textures | | iSUN | | Average | | Inference Time |
|---|---|---|---|---|---|---|---|---|---|---|---|---|---|
| | FPR95 | AUROC | FPR95 | AUROC | FPR95 | AUROC | FPR95 | AUROC | FPR95 | AUROC | FPR95 | AUROC | Per 100 Samples (s) |
| Ours | 54.90 | 89.16 | 68.75 | 86.04 | 24.54 | 94.22 | **40.21** | **91.56** | **1.10** | **99.72** | **37.10** | **92.97** | 50 |
| SONA | **3.10** | **99.39** | **44.00** | **88.35** | **18.20** | **96.19** | 58.90 | 85.20 | 63.10 | 86.17 | 37.46 | 91.06 | 160 |
| Fake it | 85.15 | 77.53 | 77.05 | 76.91 | 56.40 | 76.31 | 70.55 | 80.77 | 67.75 | 84.95 | 71.38 | 79.29 | - |

## I  FINE GRANULARITY EVALUATION ON OPENOOD V1.5

OpenOOD v1.5 (Zhang et al., 2023), as a standardized benchmark in the field of OOD detection, encompasses a variety of settings with multiple ID and OOD datasets. Among them, we have selected ImageNet-200 as the ID dataset, together with near-OOD datasets (SSB-hard, NINCO), far-OOD datasets (iNaturalist, Textures, OpenImage-O), and non-semantic OOD variants (ImageNet-V2, ImageNet-C, ImageNet-R).

### I.1  IMAGENET-200 BENCHMARK.

We applied both our model and SONA on ImageNet-100 to generate outliers, and then evaluated the resulting classifiers on ImageNet-200 (used as the ID dataset) together with near-OOD and far-OOD datasets. As shown in Table 11, our method consistently outperforms SONA across all categories, with more pronounced gains on far-OOD and covariate-shifted ID cases, demonstrating superior robustness and generalization.

We attribute this improvement to a fundamental difference in design principle: whereas SONA primarily enhances decision boundaries by training on semantically similar hard negatives, our approach focuses on generating diverse and realistic OOD samples in pixel space, thereby improving classifier resilience to a wider range of distribution shifts.

Table 11: Evaluation of diffusion-based methods on ImageNet-200 benchmark.

| Methods | Near-OOD AUROC | | | Far-OOD AUROC | | | | ID Acc |
|---|---|---|---|---|---|---|---|---|
| | SSB-hard | NINCO | Avg | iNaturalist | Textures | OpenImage-O | Avg | |
| Ours | **61.41** | **68.80** | **65.11** | **83.12** | **64.85** | **72.11** | **73.36** | **23.92** |
| SONA | 59.31 | 66.77 | 63.04 | 79.61 | 64.09 | 68.44 | 70.71 | 22.28 |

### I.2  IMAGENET-200 FULL-SPECTRUM BENCHMARK.

Additionally, we conducted the evaluation under the full-spectrum setting, which included non-semantic OOD datasets. As shown in Table 12, our method exhibits more pronounced gains over

Dream-OOD, particularly on far-OOD and non-semantic OOD benchmarks in OpenOOD v1.5, underscoring our superior robustness under diverse types of distribution shifts. We attribute this improvement to our method's ability to generate OOD samples that more faithfully reflect real-world distributional variations, thereby providing more effective supervision for classifier training.

Table 12: Evaluation of diffusion-based methods on ImageNet-200 full-spectrum benchmark.

| Methods | Near-OOD AUROC | | | Far-OOD AUROC | | | | ID Acc (Covariate-shifted) |
|---|---|---|---|---|---|---|---|---|
| | SSB-hard | NINCO | Avg | iNaturalist | Textures | OpenImage-O | Avg | |
| Ours | **48.50** | **56.45** | **52.53** | **74.36** | **52.62** | **60.70** | **62.30** | **55.12** |
| Dream-OOD | 48.12 | 56.01 | 52.06 | 73.65 | 50.88 | 56.80 | 60.44 | 51.63 |

## J  ADDITIONAL VISUAL RESULTS

### J.1  VISUALIZATION OF SYNTHETIC OOD.

In addition to the four categories presented in the main text—broccoli, candle, starfish, and strawberry, we also include ID images of the jeep category and their corresponding OOD results. Furthermore, we provide the anomaly score for each image using our proposed method.

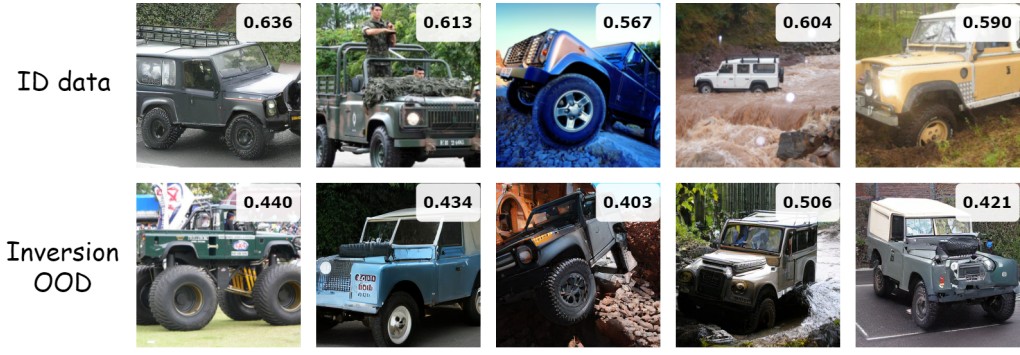

Figure 9: **ID samples and synthetic OOD of class jeep for detector training.** The severity score is provided in the upper right corner of each figure.

### J.2  VISUALIZATION OF OOD SAMPLES AS THREE TYPES.

In addition to the three categories presented in the main text—broccoli, starfish, and strawberry, we also supplement the OOD synthesis results for the candle and jeep categories, involving three types of OOD (semantic shift, covariate shift, and the ones with severe semantic and appearance changes). It should be further noted that for OOD-II and OOD-III, our zero-shot classification results are obtained based on CLIP. Specifically, we utilize the image encoder and text encoder of CLIP (Radford et al., 2021) to compute the cosine similarity between a given image and the textual descriptions of the 1,000 categories in ImageNet-1k, then assigning the category with the highest similarity as the predicted class.

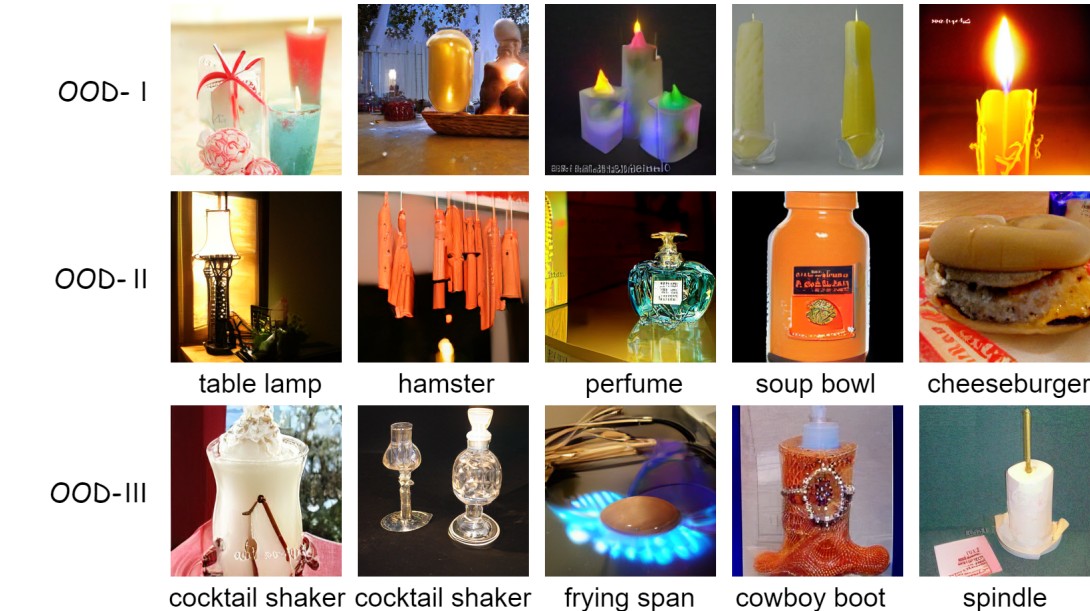

Figure 10: **OOD samples of class candle.** For OOD-II and OOD-III, the class predicted by CLIP (Radford et al., 2021) is demonstrated under each image.

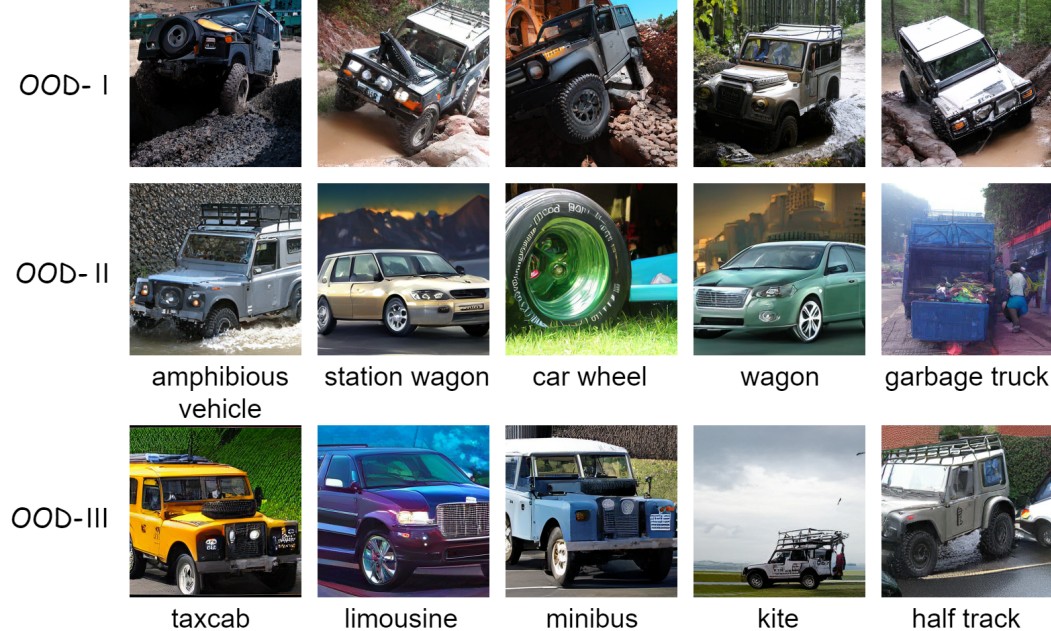

Figure 11: **OOD samples of class jeep.** For OOD-II and OOD-III, the class predicted by CLIP (Radford et al., 2021) is demonstrated under each image.

## K    DEVICES AND COMPUTATIONAL COST

We run all experiments with Python 3.8.5 and PyTorch 1.13.1, using six NVIDIA GeForce RTX 3090 GPUs.

To compare the computational performance of different methods, we conduct a comparison between Bad-OOD and other baselines, calculating the inference time of synthesizing 100 OOD samples (all experiments run on a single NVIDIA 3090 GPU). As shown in Table 13, our method is more efficient in generating OOD samples than all baselines.

Table 13: Comparison of computational cost in OOD synthesis.

| Method | Ours | VOS | NPOS | Dream-OOD | GAN | SONA |
|---|---|---|---|---|---|---|
| Inference Time Per 100 Samples (s) | **50** | 89 | 92 | 110 | 146 | 160 |

## L    THE USE OF LARGE LANGUAGE MODELS (LLMS)

In the process of writing this paper, we only employed LLMs as tools for polishing writing and retrieving relevant knowledge (e.g., finding related work).

