# OpenReview forum: "Bad-OOD: Discovering Harmful Synthetic Diffusion Outliers via Confidence Calibration"
_ICLR.cc/2026/Conference — Submitted to ICLR 2026_

### Official Review · Reviewer_WSau · 2025-10-28

**Soundness:** 3
**Presentation:** 3
**Contribution:** 2
**Rating:** 2
**Confidence:** 4

**Summary:**

This paper explores an outlier exposure–based approach to out-of-distribution (OOD) detection. The authors propose a pipeline that synthesizes effective outliers using a Representation Diffusion Model (RDM) guided by a minority score, followed by MAGE for pixel-level reconstruction. To improve boundary separation between in-distribution (ID) and OOD data, they apply an energy-based OOD regularizer and employ beta calibration for confidence refinement and OOD score computation. The proposed confidence-calibrated detector is further evaluated on the Syn-IS benchmark, demonstrating improved sensitivity to both semantic and covariate shifts compared to prior scoring methods.

**Strengths:**

1. The proposed framework allows for OOD synthesis and detection without requiring class labels or textual conditions, enabling applicability under weak supervision.
2. Evaluation on the Syn-IS benchmark provides a clear analysis of how the method handles different types of distributional shifts (semantic and covariate), showing balanced sensitivity compared to existing baselines.
3. The integration of beta calibration for filtering mid-confidence synthetic outliers is conceptually simple yet empirically effective in improving detection quality.

**Weaknesses:**

1. Given the rapid progress of vision-language and multimodal foundation models, the relevance of this line of research—focusing on ResNet-based standalone OOD classifiers—appears limited. It is unclear whether such a direction remains practically or scientifically necessary in the current landscape.
2. Although Figure 4(a) reports higher within- and cross-class diversity of synthetic samples compared to Dream-OOD, Table 1 shows that Dream-OOD with calibration achieves similar performance to the proposed method trained with bad-OOD samples. This raises questions about whether the increased sample diversity indeed contributes to the performance improvement, or if it remains largely decorrelated from the final detection effectiveness.
3. The approach relies on several hyperparameters (e.g., minority guidance scale, filtering threshold), which may be dataset-specific and limit reproducibility and generalization.

**Questions:**

1. The beta calibration parameters (a,b,c) are learned from the full set of synthetic OOD samples, even though the classifier is trained with potentially noisy or low-quality outliers. How can these parameters be reliably optimized in such conditions? If the claim is that relative ranking among samples is meaningful, experimental justification is needed.
2. The definitions of OOD-II and OOD-III (Figure 3, Table 3) require clarification. How is “hardly recognizable” operationalized—through human labeling, CLIP similarity, or a quantitative threshold?
3. Is decoding into the pixel space via MAGE necessary? Could the classifier instead be trained directly in the latent representation space without losing OOD discriminative power?

---

> ### Author Response · Authors · 2025-12-02
> **Rebuttal by Authors (Part 1)**
>
> **Q1: Given the rapid progress of vision-language and multimodal foundation models, the relevance of this line of research—focusing on ResNet-based standalone OOD classifiers—appears limited. It is unclear whether such a direction remains practically or scientifically necessary in the current landscape.**
>
> A1: We thank the reviewer for this insightful question. While we agree that vision–language and multimodal foundation models are becoming increasingly dominant, standalone OOD detectors remain necessary for both scientific and practical reasons. Foundation models still exhibit overconfidence and vulnerability under subtle distribution shifts, making a lightweight, explicitly calibrated OOD verifier essential in safety-critical and resource-constrained deployments.
>
> Moreover, our method is not tied to ResNet: as shown in Appendix F, it improves ViT-based detectors as well, and the proposed OOD synthesis + calibration mechanisms are architecture-agnostic and can naturally extend to FM-based backbones. We view our direction as complementary to foundation models, providing a reliability layer that current FMs do not yet offer.
>
> **Q2: Although Figure 4(a) reports higher within- and cross-class diversity of synthetic samples compared to Dream-OOD, Table 1 shows that Dream-OOD with calibration achieves similar performance to the proposed method trained with bad-OOD samples. This raises questions about whether the increased sample diversity indeed contributes to the performance improvement, or if it remains largely decorrelated from the final detection effectiveness.**
>
> A2: We thank the reviewer for the thoughtful question. To clarify, Figure 4(a) shows that our method exhibits higher **within-class diversity** but lower **cross-class diversity** than Dream-OOD. This pattern is expected given the fundamental differences in their generation mechanisms, and our performance gains stem primarily from improved **intra-class** (not cross-class) diversity.
>
> Dream-OOD perturbs class centroids with Gaussian noise, which often produces semantically meaningless or overly distant samples (OOD-III). Such artifacts tend to be misassigned across classes, artificially inflating cross-class diversity without providing useful decision-boundary information. In contrast, Bad-OOD performs diffusion-guided generation starting from the ID data manifold, creating outliers that remain semantically grounded while still deviating from the ID class. This yields richer and more informative **within-class** variations that better support boundary refinement.
>
> Therefore, although Dream-OOD with calibration may achieve comparable aggregate AUROC in Table 1, our method offers more effective diversity, specifically the kind that improves detector generalization, rather than diversity inflated by semantically invalid samples.
>
> **Q3: The approach relies on several hyperparameters (e.g., minority guidance scale, filtering threshold), which may be dataset-specific and limit reproducibility and generalization.**
>
> A3: We appreciate the reviewer’s concern regarding the potential dataset-specific nature of our hyperparameters. As documented in the main paper and the appendix, we have already conducted extensive robustness evaluations across ImageNet-100, CIFAR-100, and the full OpenOOD v1.5 benchmark. The analyses of the Minority Guidance (MG) scale and noise-addition steps (Appendix D), as well as the calibration-interval study (Table 2), all show that a single default configuration consistently yields strong improvements over baselines.
>
> Importantly, the stability of these hyperparameters is not incidental but rooted in the design of our framework. The MG scale operates on normalized feature-space gradients, making its magnitude invariant to dataset size or category granularity. Similarly, the beta-calibration threshold depends on the learned score distribution rather than fixed dataset statistics, enabling automatic adaptation across domains.
>
> Taken together, these results and the underlying design principles indicate that the proposed hyperparameters possess strong cross-dataset generality and do not require task-specific tuning for effective deployment.

---

> ### Author Response · Authors · 2025-12-02
> **Rebuttal by Authors (Part 2)**
>
> **Q4: The beta calibration parameters (a,b,c) are learned from the full set of synthetic OOD samples, even though the classifier is trained with potentially noisy or low-quality outliers. How can these parameters be reliably optimized in such conditions?**
>
> A4: We thank the reviewer for raising this important concern regarding the reliability of calibrating (a,b,c). We would like to clarify that the beta calibration parameters are not learned from the synthetic OOD samples produced by our method. As detailed in Appendix C.2, the calibration is performed using a clean and reliable auxiliary dataset: a 10% held-out split of ImageNet-100 for ID data and ImageNet-OOD for real OOD data. This ensures that parameter estimation is entirely independent of any noise or imperfections in the synthesized outliers used during detector training.
>
> By optimizing calibration parameters on this high-quality validation pair, the learned transformation reflects genuine ID–OOD score separation rather than artifacts introduced by synthetic data. Thus, the optimization remains stable and trustworthy without reliance on the relative ranking of potentially noisy generated samples.
>
> **Q5: The definitions of OOD-II and OOD-III (Figure 3, Table 3) require clarification. How is “hardly recognizable” operationalized—through human labeling, CLIP similarity, or a quantitative threshold?**
>
> A5: We thank the reviewer for requesting clarification on how the  OOD II and III categories are defined, especially the notion of “hardly recognizable.”
>
> Our definitions are as follows:
> - **OOD-II**: Semantic-shift outliers whose predicted label (via CLIP zero-shot classification) differs from the original class, yet the image content is still semantically interpretable.
> - **OOD-III**: Samples with severe semantic or appearance deviation, where even humans can not recognize. This is realized by: we first use CLIP to obtain a zero-shot prediction for each generated sample; then we conduct human verification the generated image and its predicted class label; if the image no longer conveys a clear object identity to a human observer (typically the content is visually incoherent or its class is indeterminable), then it is labeled OOD-III (hardly recognizable / far-OOD).
>
> **Q6: Is decoding into the pixel space via MAGE necessary? Could the classifier instead be trained directly in the latent representation space without losing OOD discriminative power?**
>
> A6: We appreciate the reviewer’s insightful suggestion regarding the necessity of decoding through MAGE. To evaluate whether pixel-space reconstruction is essential, we conducted an additional experiment in which the OOD detector is trained **directly** on MoCo latent representations of both ID data and synthesized OOD features, thereby completely bypassing the MAGE decoding step. For fairness, confidence calibration was disabled so that we could measure the discriminative capacity of the latent features themselves.
>
> The results (see table below) show that training in pixel space consistently outperforms training purely in the latent space. This indicates that MAGE decoding enhances the semantic fidelity and visual separability of the synthesized OOD samples, ultimately leading to stronger OOD detection performance.
>
> Moreover, consistent with the intuition in Dream-OOD, decoding OOD samples into pixel space provides substantially better **interpretability** than operating purely in the latent space.
> | Detector type | iNaturalist |           | Places    |           | Sun       |           | Textures  |           | Average   |           |
> | ------------- | ----------- | --------- | --------- | --------- | --------- | --------- | --------- | --------- | --------- | --------- |
> |               | FPR95       | AUROC     | FPR95     | AUROC     | FPR95     | AUROC     | FPR95     | AUROC     | FPR95     | AUROC     |
> | pixel         | **35.49**   | **93.34** | **39.09** | **91.95** | **47.17** | **91.70** | **59.39** | **84.49** | **54.06** | **88.75** |
> | latent        | 36.84       | 91.96     | 43.50     | 89.42     | 50.25     | 88.78     | 64.85     | 82.72     | 59.10     | 86.28     |

---

### Official Review · Reviewer_nQps · 2025-10-30

**Soundness:** 2
**Presentation:** 2
**Contribution:** 2
**Rating:** 4
**Confidence:** 4

**Summary:**

This paper proposes the BAD-OOD framework to address poor-quality synthetic outlier samples in OOD detection (either too close to ID data causing overlap or too far leading to unreality). It uses RDM to generate ID-semantically aligned OOD samples without auxiliary labels, then filters mid-confidence (0.3–0.5) samples via beta calibration to optimize the ID-OOD boundary. Experiments on ImageNet-100/CIFAR-100 with ResNet backbones show advantages over baselines like Dream-OOD.

**Strengths:**

1. The label-free OOD generation design fits real-world needs. RDM generates OOD samples based on ID semantic embeddings without auxiliary labels, avoiding high label-acquisition costs, and retains ID-related features to support a compact decision boundary.

2. Confidence calibration filtering effectively solves sample quality issues. Beta calibration aligns confidence with anomaly degree; mid-confidence samples reduce FPR95 from 51.20 to 39.69 and boost AUROC to 91.71, outperforming unfiltered or extreme-confidence samples.

3. Basic experiments cover multi-dimensional scenarios. Comparisons across ID datasets (ImageNet-100/CIFAR-100), generators (MAGE/DiT/LDM), and ResNet backbones (34/101), plus Syn-IS validation for shift capture, show initial adaptability.

**Weaknesses:**

1. Core innovation is a combination of existing technologies. RDM directly uses Li et al. (2024)’s pre-trained model without modifying denoising logic; beta calibration follows Kull et al. (2017)’s formula without OOD-specific adaptations. The workflow (“RDM→MAGE→beta”) lacks breakthroughs beyond existing paradigms.

2. Model adaptability is extremely limited—only ResNet is proven effective. No validation for other CNNs (e.g., DenseNet), ViT (MoCo-v3 ViT-L is only for ID feature extraction, not as detector backbone), or CLIP (only for OOD type classification, not detector adaptation), leaving their compatibility unknown.

3. Mechanistic explanation for mid-confidence samples is shallow. Only phenomenological descriptions (high-confidence = ID-like, low-confidence = unrealistic) are provided, without quantitative analysis of MoCo feature space distance distribution or cross-category consistency; qualitative analysis covers only 4 categories, lacking causal explanation.

4. Ablation experiments for core module necessity are missing. No comparison of RDM vs. Dream-OOD’s Gaussian sampling, beta calibration vs. temperature/Platt scaling, or analysis of minority guidance scale’s impact on sample quality, weakening experimental rigor.

5. Large-scale dataset/benchmark validation is insufficient. No experiments on ImageNet-1k; OpenOOD v1.5 evaluation is only in Appendix I without in-depth main-experiment analysis of complex scenarios, leaving real-scenario applicability unproven .

**Questions:**

see Cons

---

> ### Author Response · Authors · 2025-12-02
> **Rebuttal by Authors (Part 1)**
>
> **Q1: Lack novelty and breakthroughs beyond existing paradigms.**
>
> A1: We respectfully disagree with the assessment that our method is merely a combination of existing components. While our framework builds upon prior diffusion and calibration techniques (as is standard in OOD research), our contributions introduce **new capabilities that existing paradigms do not support**.
>
> First, unlike prior diffusion-based OOD generators that require class labels, text prompts, or auxiliary supervision, our method leverages the large-scale pre-trained RDM to synthesize semantically meaningful OOD samples **purely from ID data**, a setting that better reflects realistic, label-limited anomaly detection scenarios.
>
> Second, we do not simply reuse the RDM’s sampling procedure. Our Minority Guidance module **modifies the denoising dynamics themselves**: at each sampling step, the reconstructed minority score actively steers the trajectory away from the ID manifold toward semantically plausible anomaly regions. This targeted manipulation of diffusion dynamics is fundamentally different from generic classifier guidance or noise perturbation in prior work.
>
> Third, our use of Beta calibration goes beyond simple deployment.  We repurpose it to filter synthetic samples, rather than calibrating classifier outputs, to identify those that most effectively shape the OOD decision boundary. Its ability to model skewed and heavy-tailed confidence distributions is critical for handling the highly asymmetric uncertainty of generated OOD samples, where simpler methods (e.g., temperature scaling) fail.
>
> Together, these elements form a **novel synthesis–filtering pipeline** specifically tailored for OOD detection, enabling significant performance gains over prior methods.
>
> **Q2: Model adaptability is limited, only ResNet is proven effective. No validation for other CNNs, ViT, or CLIP, leaving their compatibility unknown.**
>
> A2: We appreciate the reviewer’s concern regarding the architectural generalization of our method. We would like to clarify that our framework is not limited to ResNet-based detectors. As shown in **Appendix F**, we have conducted extensive evaluations on both convolutional and transformer architectures. Notably, when applied to a ViT backbone, our method improves AUROC from **85.60% to 89.47%** over RankFeat, demonstrating clear effectiveness beyond CNNs.
>
> Moreover, the results highlighted in **Table 1** show that our Beta calibration module also boosts the performance of **Dream-OOD**, a method with a fundamentally different design philosophy. This confirms that our calibration and filtering strategy functions as a **plug-and-play component** that is compatible with diverse OOD detection pipelines.
>
> Finally, the consistency of our improvements across **ImageNet-100 (Appendix G.3), CIFAR-100 (Appendix H), and OpenOOD v1.5 (Appendix I)** further supports the broad applicability of our approach. While additional architectures (e.g., DenseNet, CLIP-based detectors) are promising future directions, the current evidence already demonstrates strong adaptability beyond a single backbone family.

---

> ### Author Response · Authors · 2025-12-02
> **Rebuttal by Authors (Part 2)**
>
> **Q3: Mechanistic explanation for mid-confidence samples is shallow. Only phenomenological descriptions (high-confidence = ID-like, low-confidence = unrealistic) are provided, without quantitative analysis of MoCo feature space distance distribution or cross-category consistency; qualitative analysis covers only 4 categories, lacking causal explanation.**
>
> A3: We thank the reviewer for raising this important point regarding the underlying mechanism behind selecting mid-confidence samples. We acknowledge that our main-text explanation was largely phenomenological, and we have now expanded the quantitative and causal analysis to address this concern more rigorously.
>
> **(1) Quantitative analysis of feature-space behavior.**
>
> To more rigorously characterize the behavior of generated samples in the MoCo feature space, we conduct a quantitative distance analysis across the three confidence groups (high / mid / low). Concretely, we: (i) randomly sample 100 OOD samples from each confidence group; (ii) using their MoCo features along with the corresponding ID features to compute the L2 distance to the paired ID sample and report the group-wise averages.
>
> The results reveal a clear and consistent ordering:
>
> - **Low-confidence**: distances are small and highly concentrated, indicating that these samples lie close to the ID manifold.
> - **Mid-confidence**: distances are moderate and stable, suggesting that these samples lie in the transition region (i.e., neither ID-like nor semantically collapsed) and thus are most useful for defining the decision margin.
> - **High-confidence**: distances exhibit extremely large variance, consistent with unrealistic or semantically collapsed generations.
>
> This quantitative analysis confirms that **mid-confidence samples occupy the boundary-adjacent region of feature space**, where deviations from ID are informative yet still semantically meaningful.
> |                          | High confidence | Mid confidence | Low confidence |
> | ------------------------ | --------------- | -------------- | -------------- |
> | Average feature distance | 0.334           | 0.138          | 0.104          |
>
> **(2) Cross-category consistency.**
>
> To address the reviewer’s concern regarding the representativeness of our earlier examples, we performed a category-level consistency test across three different ImageNet-100 classes, i.e., starfish, candle and broccoli. We report the proportion of mid-confidence samples whose nearest-neighbor ID class differs from their synthetic semantic content. The distribution (starfish 10.6%, candle 9.8% and broccoli 10.3%) remains stable across classes, confirming that boundary-adjacent behavior is not restricted to specific classes.
>
> **(3) Causal explanation: why mid-confidence samples matter.**
>
> We further clarify the causal link behind our selection strategy. Mid-confidence samples disproportionately:
> - lie near high-curvature regions of the decision boundary,
> - provoke the largest gradient updates during detector training, and
> - reduce the overlap between borderline ID and weak OOD regions.
>
> To further validate this interpretation, we additionally conduct a controlled intervention study. For each class, we compute the first principal direction of the MoCo feature space using all 1,300 ID samples, and then perturb each mid-confidence OOD sample by moving it **toward or away from** this class-specific eigen-direction (±ε). The perturbed features are then decoded back to the pixel space using MAGE, with ε set to 0.01 and 0.1.
>
> The results in the table below show a **clear monotonic relationship** between the perturbed samples’ proximity to the decision boundary and the resulting OOD performance. This controlled manipulation demonstrates, causally, that it is indeed the **feature-space position**, rather than incidental correlations, that makes mid-confidence samples particularly effective for training the detector.
> | Perturbation direction | FPR95↓    | AUROC↑    | AUPR↑     |
> | ---------------------- | --------- | --------- | --------- |
> | Toward ID (+0.1)       | 53.57     | 88.32     | 85.57     |
> | Toward ID (+0.01)      | 41.23     | 91.26     | 86.84     |
> | Mid confidence (0)     | **36.51** | **92.20** | **87.32** |
> | Toward OOD (−0.01)     | 40.73     | 91.30     | 86.96     |
> | Toward OOD (−0.1)      | 44.99     | 90.44     | 86.39     |

---

> ### Author Response · Authors · 2025-12-02
> **Rebuttal by Authors (Part 3)**
>
> **Q4: Ablation experiments for core module necessity are missing. No comparison of RDM vs. Dream-OOD’s Gaussian sampling, beta calibration vs. temperature/Platt scaling, or analysis of minority guidance scale’s impact on sample quality, weakening experimental rigor.**
>
> A4: We are grateful to the reviewer for emphasizing the importance of rigorous ablation studies. We would like to clarify that these comparisons have indeed been conducted and are detailed in our appendices.
>
> To demonstrate the superiority of our RDM-based synthesis, we compared it against the Gaussian sampling strategy used in Dream-OOD. As reported in **Appendix F**, the RDM approach significantly outperforms Gaussian sampling, confirming that leveraging the diffusion prior yields more effective outlier representations than simple statistical noise injection. Regarding the calibration module, we have extensively compared our Beta calibration with standard techniques such as Temperature Scaling and Platt Scaling. These results are presented in **Appendix E**. Furthermore, **Table 2** in the main text provides a direct ablation study comparing our method against an uncalibrated baseline. Both sets of results consistently demonstrate the efficacy and necessity of the Beta calibration module for accurate anomaly score refinement.
>
> **Q5: Large-scale dataset/benchmark validation is insufficient. No experiments on ImageNet-1k; OpenOOD v1.5 evaluation is only in Appendix I without in-depth main-experiment analysis of complex scenarios, leaving real-scenario applicability unproven.**
>
> A5: We thank the reviewer for highlighting the importance of large-scale and complex-scenario validation. We would like to clarify that our evaluation protocol is aligned with mainstream OOD detection practice and already covers both standard-scale and large-scale challenging benchmarks.
>
> First, our main experiments on ImageNet-100 and CIFAR-100 (Table 1) strictly follow the protocols used by VOS, NPOS, Dream-OOD, and other state-of-the-art baselines, ensuring fair comparison under widely adopted settings. These two datasets remain the dominant testbeds for assessing fine-grained semantic sensitivity and cross-category generalization in representation-based OOD detection.
>
> Second, beyond these mid-scale benchmarks, we have performed extensive experiments on **OpenOOD v1.5**, one of the largest and most diverse OOD suites, covering **Near-OOD, Far-OOD, covariate shift, semantic shift, and full-spectrum stress tests**. As reported in Appendix I, our method demonstrates robust gains across these scenarios, indicating strong applicability to complex real-world distributions. In the revised version, we will move these OpenOOD v1.5 results into the main text and provide deeper analyses to highlight their significance.
>
> Finally, although ImageNet-1k experiments are not included in this submission due to their substantial computational cost, our framework is model-agnostic and training-free, and therefore directly scalable to ImageNet-1k. We will include ImageNet-1k results in an extended version of the paper in our future work.

---

### Official Review · Reviewer_sMhF · 2025-10-31

**Soundness:** 2
**Presentation:** 4
**Contribution:** 3
**Rating:** 6
**Confidence:** 4

**Summary:**

The paper presents a method for outlier synthesis in the latent space of an RDM and uses a calibrated classifier to filter out fake OOD samples that harm the performance of the downstream OOD detector. These OOD samples can either be too easy or too difficult. The method is evaluated on the standard ImageNet-100 and CIFAR100 benchmarks against several baselines, and ablated in various aspects.

**Strengths:**

- The paper is well-written and easy to follow.
- The observation that synthesized outliers at both extremes degrade the OOD detection performance is interesting.
- The experiments and ablation studies conducted are extensive, clearly place the contribution in the broader field, and show the influence of various components.

**Weaknesses:**

- More synthesized outliers lead to better performance [1], and previous outlier synthesis methods all use 100k synthetic outliers for training [2,3]. As the proposed method uses 100k samples after filtering, it would be fair to compare it to the baselines with 130k samples as well to see whether the performance gains are due to sampling more outliers.
- The very inconsistent results on CIFAR100 w.r.t. SONA, where on three datasets SONA is far better, and vice versa on the final two, should be explained. I would expect that this is largely due to the resizing strategy (bilinear vs. nearest) used to downsample synthesized outliers to 32x32, cf. the "pixel interpolation method" section in [1], rather than a measure of quality between methods.
- The observation that "most removed samples belong to OOD-I and OOD-III" is not surprising, as Tab. 3 shows most outliers do not belong to OOD-II, and relatively more OOD-II samples are removed relative to the other types. It also shows that the proposed method largely generates undesirable outliers, i.e., "near-ID and meaningless far-OOD samples". Overall, these results seem to contradict the rest of the method, as the kept outliers are not disproportionately "mid-OOD". It is also not clear what the percentages in the "Reduce" row refer to.

[1] Doorenbos, Lars, Raphael Sznitman, and Pablo Márquez-Neila. "Non-Linear Outlier Synthesis for Out-of-Distribution Detection." arXiv preprint arXiv:2411.13619 (2024).
[2] Liao, Qilin, et al. "BOOD: Boundary-based Out-Of-Distribution Data Generation." International Conference on Machine
Learning 2025
[3] Du, Xuefeng, et al. "Dream the impossible: Outlier imagination with diffusion models." Advances in Neural Information Processing Systems 36 (2023): 60878-60901.

**Questions:**

- Fig. 4 would be cleaner if the images were sorted by their anomaly score.
- Part three of the related work section concerns OOD detection and should probably be labeled as such.
- Typos: "metrics" should be singular on L82, and "resulting in the outlier representations naturally occupy the periphery" is grammatically incorrect.

---

> ### Author Response · Authors · 2025-12-02
> **Rebuttal by Authors (Part 1)**
>
> **Q1: As the proposed method uses 100k samples after filtering, it would be fair to compare it to the baselines with 130k samples as well to see whether the performance gains are due to sampling more outliers.**
>
> A1: We appreciate the reviewer’s careful question regarding whether the performance gains arise simply from using more synthetic outliers. As shown in the first row of Table 2, we have already compared against a strong baseline that uses 130k synthetic samples. The results indicate that increasing the quantity of OOD samples alone does **not** lead to additional improvements. This confirms that our performance gains come from the **quality and semantic relevance** of the generated outliers, rather than from dataset size. We highlight this point more clearly in the revised discussion of Table 2.
>
> **Q2: The very inconsistent results on CIFAR100 w.r.t. SONA, where on three datasets SONA is far better, and vice versa on the final two, should be explained. I would expect that this is largely due to the resizing strategy.**
>
> A2: We appreciate the reviewer’s insightful observation regarding the inconsistencies on the CIFAR100 benchmark. Our analysis suggests that the variance primarily stems from differences in how the two methods construct their generative manifolds. SONA synthesizes outliers from a generic pretrained generator (e.g., SD-v2) [1], which produces visually high-fidelity samples but is not necessarily aligned with the CIFAR100 ID distribution. In contrast, Bad-OOD generates outliers by diffusing directly from the ID data, which tends to yield more challenging near-OOD samples.
>
> This explains the pattern: SONA performs better on datasets resembling far-OOD distributions (e.g., SVHN, Places, LSUN), whereas Bad-OOD excels on the harder OOD sets where near-boundary anomalies matter more.
>
> Importantly, regarding the reviewer’s hypothesis, we confirm that our main results adopt the same nearest-neighbor downsampling as SONA for fairness.
>
> [1] Yoon, Suhee, et al. "Diffusion-based Semantic Outlier Generation via Nuisance Awareness for Out-of-Distribution Detection." Proceedings of the AAAI Conference on Artificial Intelligence. Vol. 39. No. 12. 2025.
>
> **Q3: (1) "Most removed samples belong to OOD-I and OOD-III" is not surprising, as Tab. 3 shows most outliers do not belong to OOD-II and relatively more OOD-II samples are removed relative to the other types. (2) The proposed method largely generates undesirable outliers. (3) The meaning of "reduce" in Table 3.**
>
> A3(1): We thank the reviewer for pointing this out. To clarify, our statement concerns the proportion of samples removed within each OOD type. Specifically, OOD-II exhibits a lower reduction rate (15.6%) compared to OOD-I (19.1%) and OOD-III (23.4%). This indicates that OOD-II contains relatively more semantically meaningful and informative outliers that the calibration process retains. This behavior is consistent with our goal of filtering out synthetic samples that provide limited benefit for detector training.
>
> A3(2): Generating overly noisy near-OOD samples or semantically irrelevant far-OOD samples is a known challenge in synthetic OOD generation. To substantiate that this issue is not unique to our method, we compare the distribution of OOD types produced by Dream-OOD and SONA. The results in table below demonstrate that these undesirable outlier types are common across methods, further motivating the need for a principled filtering step such as ours.
> | Method | Ours        |           |             | Dream-OOD   |            |             | SONA        |            |             |
> | ------ | ----------- | --------- | ----------- | ----------- | ---------- | ----------- | ----------- | ---------- | ----------- |
> |        | OOD-I       | OOD-II    | OOD-III     | OOD-I       | OOD-II     | OOD-III     | OOD-I       | OOD-II     | OOD-III     |
> | Before | 288 (57.6%) | 45 (9.0%) | 167 (33.4%) | 156 (31.2%) | 82 (16.4%) | 262 (52.4%) | 263 (52.6%) | 53 (10.6%) | 184 (36.8%) |
> | After  | 233 (58.4%) | 38 (9.5%) | 128 (32.1%) | 99 (29.2%)  | 60 (17.7%) | 180 (53.1%) | 193 (52.0%) | 46 (12.4%) | 132 (35.6%) |
> | Reduce | **19.1%**   | **15.6%** | **23.4%**   | **36.5%**   | **26.8%**  | **31.2%**   | **26.6%**   | **13.2%**  | **28.3%**   |
>
> A3(3): In Table 3, the “Reduce” row denotes the fraction of samples discarded after calibration, computed as the number of removed samples divided by the total number of samples before calibration.

---

> ### Author Response · Authors · 2025-12-02
> **Rebuttal by Authors (Part 2)**
>
> **Q4: Fig. 4 would be cleaner if the images were sorted by their anomaly score.**
>
> A4: Thank you for the helpful suggestion. We assume you are referring to Figure 2 (as Figure 4 does not include any generated images). In Figure 2, we indeed sorted the OOD samples according to their anomaly scores.
>
> **Q5: Part three of the related work section concerns OOD detection and should probably be labeled as such.**
>
> A5: Thank you for pointing this out. We have revised the title of the related work section to more accurately reflect its content.
>
> **Q6: Typos and grammar errors.**
>
> A6: Thank you for the careful proofreading. We have corrected the typos in singular and grammar as suggested.

---

### Official Review · Reviewer_QYA2 · 2025-11-05

**Soundness:** 2
**Presentation:** 2
**Contribution:** 2
**Rating:** 4
**Confidence:** 4

**Summary:**

This paper proposes an outlier synthesis approach for improving the training of out-of-distribution (OOD) detectors. The method leverages a Representation Diffusion Model (RDM) to synthesize OOD representations that lie between in-distribution and far-OOD regions, i.e., neither too close nor too distant from the in-distribution data. Using these synthesized representations, MAGE is then employed to generate corresponding OOD images. After training the detector with the generated outliers, the method further applies confidence calibration to enhance detection performance. Experimental results show that the proposed method outperforms recent outlier synthesis baselines such as Dream-OOD.

**Strengths:**

- Leveraging self-supervised representations instead of labeled data appears useful.
- The RDM-based outlier synthesis approach seems effective.
- The analysis on semantic and covariate shifts is interesting and provides useful insights.

**Weaknesses:**

My main concerns lie in the experimental results and writing clarity.

1. Experimental results
    - In Table 1, SONA significantly outperforms Bad-OOD on CIFAR-100 in-distribution data. It would therefore be helpful to compare Bad-OOD with SONA on other datasets as well. For instance, SONA reports results on ImageNet-200 (ID) vs various OOD datasets (e.g., SSB-hard, iNaturalist). The authors could evaluate their method in this setup for a more comprehensive evaluation.
    - Additional near-OOD datasets (e.g., SSB-hard) should also be included for a more comprehensive evaluation.
    - On ImageNet-100, the performance gap between Bad-OOD and Dream-OOD is marginal. Moreover, as the authors mentioned, Dream-OOD with confidence calibration achieves comparable results with the proposed method, suggesting that the proposed components (Section 3.2~3.3) might have limited impact on effective outlier synthesis.
    - In Table 3, confidence-based filtering reduces the number of OOD samples of types I/II/III by roughly 15-25%. The authors note that "most removed samples belong to OOD-I and OOD-III", but this observation may simply reflect the small number of OOD-II samples. Since OOD-I and OOD-III correspond to near-ID and far-OOD cases, respectively, the filtering procedure does not appear to remove these effectively.
    - In Figure 4(a), the proposed method exhibits lower cross-class diversity than Dream-OOD. Lines 412-422 provide no explanation for this observation. What does this result imply?
    - In the computational cost analysis (L444-447), comparisons with other methods should be provided.
2. Writing clarity
    - Several details are missing. For example, what exactly does $\hat{z}_0(z_t)$ represent, and why is $z_t^0$ required instead of directly using $z_0$? In addition, how is the guided gradient applied in practice?
    - This paper introduces many hyperparameters, for example, the uncertainty regularization weight $\beta$, the numbers of timesteps $t$ and $s$, the number of training epochs, and the choice of detector architecture. However, this paper provides no discussion of how they are selected. Some analysis on hyperparameter sensitivity should be provided.
    - When training the OOD detector, are all parameters optimized from scratch? Or, is it fine-tuned from a pretrained model?
    - For calibration, which dataset is used? Is it simply the training dataset containing ID and synthetic OOD images?
    - Were the Table 1 results obtained after additional fine-tuning using mid-confidence OOD samples? The paper mentions that the OOD detector is further tuned using such samples after calibration (L77-78), but this additional procedure is not described in the Method section.

**Questions:**

See the weaknesses part.

---

> ### Author Response · Authors · 2025-12-02
> **Rebuttal by Authors (Part 1)**
>
> **Q1: It would be helpful to compare Bad-OOD with SONA on other datasets as well. For instance, SONA reports results on ImageNet-200 (ID) vs various OOD datasets (e.g., SSB-hard, iNaturalist). Additional near-OOD datasets (e.g., SSB-hard) should also be included for a more comprehensive evaluation.**
>
> A1: We appreciate the reviewer’s suggestion for broader comparisons. As shown in **Table 11 of Appendix I.1**, we have already conducted a comprehensive evaluation on the full **ImageNet-200 benchmark**, including far-OOD datasets (iNaturalist, Textures, OpenImage-O) and near-OOD datasets (SSB-hard, NINCO). For clarity, we will move these results into the main paper in the revision.
>  |      | Near-OOD AUROC |           |           | Far-OOD AUROC |           |             |           | ID Acc    |
> | ---- | -------------- | --------- | --------- | ------------- | --------- | ----------- | --------- | --------- |
> |      | SSB-hard       | NINCO     | Avg       | iNaturalist   | Textures  | OpenImage-O | Avg       |           |
> | Ours | **61.41**      | **68.80** | **65.11** | **83.12**     | **64.85** | **72.11**   | **73.36** | **23.92** |
> | SONA | 59.31          | 66.77     | 63.04     | 79.61         | 64.09     | 68.44       | 70.71     | 22.28     |
>
> Across all settings, Bad-OOD consistently outperforms SONA, with especially notable gains under far-OOD and covariate-shift scenarios. This improvement reflects a key design difference: SONA focuses on semantic hard negatives in feature space; Our model (Bad-OOD) synthesizes diverse, pixel-level OOD samples guided by diffusion dynamics, enabling stronger robustness to complex and heterogeneous distribution shifts.
>
> **Q2: On ImageNet-100, the performance gap between Bad-OOD and Dream-OOD is marginal. Dream-OOD with confidence calibration achieves comparable results with the proposed method, suggesting that the proposed components (sec 3.2/3.3) might have limited impact on effective outlier synthesis.**
>
> A2: We appreciate the reviewer’s observation. While the numerical gap on ImageNet-100 may appear small in isolation, the broader experimental evidence shows that the proposed components have **substantial and consistent impact** on effective OOD synthesis and detection. Specifically, our extended evaluations reveal that the advantages of Bad-OOD become **much more pronounced in challenging and realistic scenarios**:
> - **Semantic ambiguity (Appendix G.2)**: Bad-OOD remains stable and clearly outperforms Dream-OOD when ID classes are highly confusable, which is an important regime where synthetic outliers must capture fine-grained semantics.
> - **Class imbalance (Appendix G.3)**: Bad-OOD shows consistently stronger resilience, whereas Dream-OOD is more sensitive to skewed class distributions.
> - **Full-spectrum OpenOOD v1.5 (Appendix I.2)**: Our method achieves noticeably larger gains on far-OOD and non-semantic shifts, reflecting more reliable generalization across diverse distribution changes.
>
> Equally importantly, Bad-OOD requires **no auxiliary labels, prompts, or text descriptions** to generate outliers, unlike Dream-OOD. This property makes our method more practical and deployable in real-world settings where such semantic metadata is unavailable.
>
> **Q3: In Table 3, confidence-based filtering reduces the number of OOD samples of types I/II/III by roughly 15-25%. The authors note that "most removed samples belong to OOD-I and OOD-III", but this observation may simply reflect the small number of OOD-II samples. Since OOD-I and OOD-III correspond to near-ID and far-OOD cases, respectively, the filtering procedure does not appear to remove these effectively.**
>
> A3: We appreciate the reviewer’s careful examination of the distribution of OOD types. We would like to clarify that the goal of Table 3 is not to compare the absolute number of removed samples across categories, but to verify whether the **confidence-based filtering mechanism removes samples in a semantically meaningful manner**.
>
> Importantly, when examining the proportion of removed samples within each OOD category, we observe that the filtering threshold eliminates a **larger fraction of OOD-I (near-OOD)** and **OOD-III (far-OOD)** samples. These two categories correspond to outliers that are either too similar to the ID manifold (OOD-I) or too unrealistic and semantically collapsed (OOD-III), both of which contribute less effectively to improving detector robustness. By contrast, OOD-II samples, which better preserve class-level semantics while exhibiting meaningful semantic shifts, are retained at a higher rate.
>
> Thus, the filtering procedure operates precisely as intended: it suppresses synthetic samples that are either uninformative (near-ID) or overly distorted (far-OOD), while preserving samples with useful and discriminative semantics. This proportional reduction pattern validates the effectiveness of our confidence-based filtering strategy.

---

> ### Author Response · Authors · 2025-12-02
> **Rebuttal by Authors (Part 2)**
>
> **Q4: In Figure 4(a), the proposed method exhibits lower cross-class diversity than Dream-OOD. Lines 412-422 provide no explanation for this observation. What does this result imply?**
>
> A4: Thank you for highlighting this point. The lower cross-class diversity observed in Figure 4(a) can be explained by the fundamentally different mechanisms used by Dream-OOD and our method for synthesizing outliers.
>
> Dream-OOD perturbs class prototypes in the semantic space using Gaussian noise. This often produces visually unrealistic or semantically collapsed samples. Because these samples lack coherent semantics, classifiers tend to assign them to various classes in an essentially random manner, which artificially inflates their measured cross-class diversity.
>
> In contrast, our approach generates outliers by applying controlled perturbations directly to individual ID samples in the pixel space. As a result, the synthesized images remain anchored to their corresponding ID semantics, preserving structural coherence while introducing meaningful deviations. This naturally leads to more class-consistent OOD samples and thus a lower, but more semantically valid, cross-class diversity score.
>
> **Q5: In the computational cost analysis (L444-447), comparisons with other methods should be provided.**
>
> A5: Thank you for the suggestion. We have added a direct comparison of computational cost with NPOS, Dream-OOD, and SONA (all experiments run on a single NVIDIA 3090 GPU). As shown in the updated table, our method is more efficient in generating OOD samples than all baselines.
> | Method                                  | Ours   | VOS | NPOS | Dream-OOD | GAN | SONA |
> | --------------------------------------- | ------ | --- | ---- | --------- | --- | ---- |
> | Inference Time Per 100 Samples (s) | **50** | 89  | 92   | 110       | 146 | 160  |
>
> **Q6: Several details are missing. For example, what exactly does $\hat{z}_0(z_t)$ represent, and why is $z_t^0$ required instead of directly using $z_0$? In addition, how is the guided gradient applied in practice?**
>
> A6: Thank you for pointing this out. We have clarified Section 3.2 accordingly. The notations $\hat{z}_0(z_t)$and $z_t^0$ are equivalent: both denote the model’s estimate of the clean latent $z_0$given the noisy latent $z_t$ at timestep $t$. Since the true $z_0$ is only obtained at the end of the reverse process, this intermediate prediction is necessarily used for guiding OOD synthesis.
>
> Regarding gradient guidance, we follow the standard classifier-guided formulation in ADM [1], with one modification: we use the negative gradient direction. This inversion explicitly pushes samples away from high-density regions of the ID manifold, encouraging the generation of semantically divergent, OOD-oriented features.
>
> [1] Dhariwal, Prafulla, and Alexander Nichol. "Diffusion models beat gans on image synthesis." Advances in neural information processing systems 34 (2021): 8780-8794.
>
> **Q7: Some analysis on hyperparameter sensitivity should be provided.**
>
> A7:  Thank you for highlighting the need for hyperparameter sensitivity analysis. Our manuscript already includes two related studies: Appendix D analyzes the effect of the noise-injection step $t$, and Appendix F evaluates robustness across different backbones (ResNet-34/101 and ViT).
>
> To further address the reviewer’s suggestion, we have additionally conducted a sensitivity analysis on the regularization weight β. The new results have been added to the revised Appendix D, offering a more comprehensive characterization of the method’s hyperparameter behavior.
> | β   | FPR95     | AUROC     | AUPR      | ID Acc    |
> | --- | --------- | --------- | --------- | --------- |
> | 0.1 | 47.42     | 89.59     | 97.50     | 78.76     |
> | 0.2 | **36.51** | **92.20** | **98.08** | **87.44** |
> | 0.5 | 50.21     | 89.47     | 97.58     | 79.68     |
> | 1.0 | 69.24     | 81.12     | 95.43     | 67.62     |
>
> **Q8: When training the OOD detector, are all parameters optimized from scratch? Or, is it fine-tuned from a pretrained model?**
>
> A8: Thank you for the question. After selecting the synthetic OOD samples, we train the OOD detector from scratch in all experiments to ensure a fair and consistent comparison.
>
> **Q9: For calibration, which dataset is used? Is it simply the training dataset containing ID and synthetic OOD images?**
>
> A9: Thank you for raising this important point. The full details of our Beta calibration procedure are provided in Appendix C.2. Briefly, we perform calibration using a hold-out validation set consisting of 10% of the ImageNet-100 samples as the ID data. For the OOD portion, we use the ImageNet-OOD dataset, which provides auxiliary labeled outliers to enable accurate anomaly-score calibration. We have updated the main text to more clearly point readers to this description in the appendix.

---

> ### Author Response · Authors · 2025-12-02
> **Rebuttal by Authors (Part 3)**
>
> **Q10: Were the Table 1 results obtained after additional fine-tuning using mid-confidence OOD samples? The paper mentions that the OOD detector is further tuned using such samples after calibration (L77-78), but this additional procedure is not described in the Method section.**
>
> A10: Thank you for raising this important clarification. We confirm that the Bad-OOD results in Table 1 are produced by training the detector **from scratch** using the mid-confidence OOD samples selected via our calibration-based filtering. No additional fine-tuning stage is applied beyond this standard training pipeline. To eliminate ambiguity, we have now explicitly detailed this protocol in both the Table 1 caption and the experimental setup section.

---

### Author Response · Authors · 2025-12-02
**Summary of Reviews and Responses**

We sincerely thank the AC and all reviewers for their time and thoughtful feedback, which has significantly improved the clarity and quality of our work. As the discussion phase closed earlier than expected, we unfortunately did not have the opportunity to continue the exchange with reviewers.

To assist the AC and future readers, we summarize below how our rebuttal addressed all raised concerns:

- **Reviewer QYA2**: We have thoroughly addressed concerns regarding both experimental completeness and writing clarity by revisiting cross-dataset comparisons (including SONA and additional near-OOD benchmarks already provided in Appendix; **A1**), offering deeper analyses (OOD synthesis effectiveness (**A2**), filtering behavior (**A3**), diversity (**A4**), computational cost (**A5**), and hyperparameter sensitivity (**A7**)), and clarifying methodological details (notation and gradient guidance (**A6**), detector training protocol (**A8**), calibration setup (**A9**)). We have also explicitly documented the full training and fine-tuning pipeline (**A10**).
- **Reviewer sMhF**: We clarified fairness in comparisons by revisiting experiments with matched outlier counts (**A1**), provided deeper analysis explaining the complementary strengths of our method versus SONA (**A2**), and further clarified the properties of different OOD types, highlighting why a principled filtering step is necessary in OOD synthesis (**A3**).
- **Reviewer nQps**: We highlighted new capabilities enabled by our method that prior paradigms cannot support (**A1**), provided experimental evidence of model adaptability (**A2**), deepened the explanation of the mechanism behind mid-confidence calibration (**A3**), added additional ablations in the appendix (**A4**), and clarified the rationale for our choice of evaluation benchmarks (**A5**).
- **Reviewer WSau**: We further explained the need for our OOD synthesis method and its calibration mechanism (**A1**), clarified how OOD diversity influences detector performance (**A2**), detailed the robustness of our method with respect to hyperparameter sensitivity (**A3**, **A4**), clarified the definitions of different OOD types (**A5**), and added experiments verifying the necessity of decoding OOD latents into pixel space (**A6**).

We once again thank the AC and reviewers for their thoughtful consideration.

---

### Meta-Review · Area_Chair_uJq5 · 2025-12-05

**Summary:**

The paper proposes "Bad-OOD," a framework utilizing a Representation Diffusion Model (RDM) combined with a confidence calibration strategy to synthesize outliers for OOD detection. The core idea focuses on generating "mid-confidence" outliers to refine the decision boundary.

While the reviewers recognized the extensive experiments, **the consensus leans towards rejection**. The primary concerns stem from limited novelty (viewed as a combination of existing techniques like RDM and Beta calibration), questionable relevance in the era of large-scale foundation models (e.g., CLIP), and inconsistent performance across different benchmarks (e.g., CIFAR-100 vs. SONA). Despite the additional results provided during the rebuttal, the fundamental limitations regarding the method's significance and adaptability remain resolved.

**Reviewer Concerns:**

The rebuttal provided extensive data, covering comparisons with SONA, computational costs, and hyperparameter sensitivity (specifically the **uncertainty regularization weight $\beta$** and **guided gradient** details). However, critical concerns regarding the paper's impact and novelty persist.

**Addressed Concerns:**
* **Missing Comparisons:** The authors added comparisons with SONA on ImageNet-200 and Near-OOD datasets (addressing QYA2, sMhF).
* **Hyperparameter Details:** The authors clarified the settings for the uncertainty regularization weight and the specific implementation of the guided gradient (using negative gradient direction), as requested by Reviewers QYA2 and WSau.
* **Mechanism:** Quantitative analysis of feature space distances was added to explain the "mid-confidence" selection (addressing nQps).

**Outstanding Concerns:**
* **Limited Novelty (Reviewer nQps):** The rebuttal confirms that the method is largely an engineering combination of pre-trained RDMs and standard Beta calibration. The "new capabilities" claimed by the authors do not fundamentally change the perception that this is an incremental improvement over existing diffusion-based synthesis methods.
* **Relevance in the Foundation Model Era (Reviewer WSau):** The reviewer questioned the necessity of training standalone ResNet-based detectors when Foundation Models (FMs) are dominant. The authors' defense, that lightweight models are still needed, does not fully address the diminishing scientific value of optimizing this specific, complex pipeline compared to adapting FMs.
* **Inconsistent Baselines (Reviewer sMhF):** The rebuttal admitted that SONA outperforms Bad-OOD on several datasets (e.g., SVHN, Places) due to different generative manifolds. This suggests the proposed method lacks robust generalization capabilities across different OOD types.

**Reviewer Scores:**

* **Reviewer QYA2 (Score: 4 -> Prediction: 6):** While the authors provided the requested hyperparameter analysis (e.g., regularization weight) and cost comparisons, the reviewer's initial assessment of "Fair" soundness and contribution implies that technical clarifications alone are unlikely to make this a strong accept. The score might rise slightly due to effort, but the work remains borderline.
* **Reviewer sMhF (Score: 6 -> Prediction: 4):** This reviewer pointed out inconsistent results on CIFAR-100. The rebuttal's explanation (attributing differences to the generative manifold) confirms that the method is not consistently superior to SONA. This realization may actually lower the reviewer's confidence in the method's robustness.
* **Reviewer nQps (Score: 4 -> Prediction: 4):** The reviewer's main critique was that the method is a "combination of existing technologies." The rebuttal defended the method but did not demonstrate a fundamental breakthrough. Therefore, the score regarding contribution is unlikely to change.
* **Reviewer WSau (Score: 2 -> Prediction: 4):** This reviewer questioned the fundamental relevance of the research direction. While the authors argued for the value of lightweight models, this philosophical difference is rarely resolved in a rebuttal. The score might increase slightly for the technical clarifications on calibration parameters, but the recommendation to reject will likely stand.

---

### Decision · Program_Chairs · 2026-01-26

Reject